# Efficient RL with Impaired Observability: Learning to Act with Delayed and Missing State Observations

**Minshuo Chen**[1]    **Yu Bai**[2]    **H. Vincent Poor**[1]    **Mengdi Wang**[1]
[1]Princeton University    [2]Salesforce Research

## Abstract

In real-world reinforcement learning (RL) systems, various forms of *impaired observability* can complicate matters. These situations arise when an agent is unable to observe the most recent state of the system due to latency or lossy channels, yet the agent must still make real-time decisions. This paper introduces a theoretical investigation into efficient RL in control systems where agents must act with delayed and missing state observations. We establish near-optimal regret bounds, of the form $\widetilde{\mathcal{O}}(\sqrt{\text{poly}(H)SAK})$, for RL in both the delayed and missing observation settings. Despite impaired observability posing significant challenges to the policy class and planning, our results demonstrate that learning remains efficient, with the regret bound optimally depending on the state-action size of the original system. Additionally, we provide a characterization of the performance of the optimal policy under impaired observability, comparing it to the optimal value obtained with full observability.

## 1 Introduction

In Reinforcement Learning (RL), an agent engages with an environment in a sequential manner. In an ideal setting, at each time step, the agent would observe the current state of the environment, select an action to perform, and receive a reward Smallwood and Sondik [1973], Bertsekas [2012], Sutton and Barto [2018], Lattimore and Szepesvári [2020]. However, real-world engineering systems often introduce impaired observability and latency, where the agent may not have immediate access to the instant state and reward information. In systems with lossy communication channels, certain state observations may even be permanently missing, never reaching the agent. Nevertheless, the agent is still required to make real-time decisions based on the available information.

The presence of impaired observability transforms the system into a complex interactive decision process (Figure 1), presenting challenges for both learning and planning in RL. With limited knowledge about recent states and rewards, the agent's policy must extract information from the observed history and utilize it to make immediate decisions. This introduces significant complexity to the policy class and poses difficulties for RL. Moreover, the loss of information due to permanently missing observations further hampers the efficiency of RL methods. Although a naïve approach would involve augmenting the state and action space to create a fully observable Markov Decision Process (MDP), such a method would lead to exponential regret growth in the state-action size.

**Why existing methods do not work.** One may be tempted to cast the problem of impaired observability into a Partially Observed MDPs (POMDPs). However, this would not solve the problem. In POMDP, the system does not reveal its instant state to the agent but provides an emission state observation conditioned on the latent state. POMDPs are known to suffer from the curse of history Papadimitriou and Tsitsiklis [1987], Bertsekas [2012], Krishnamurthy [2016], unless additional assumptions are imposed. Existing efficient algorithms focus on subclasses of POMDPs with decodable or distinguishable partial observations Jin et al. [2020], Uehara et al. [2022], Zhan et al.

37th Conference on Neural Information Processing Systems (NeurIPS 2023).

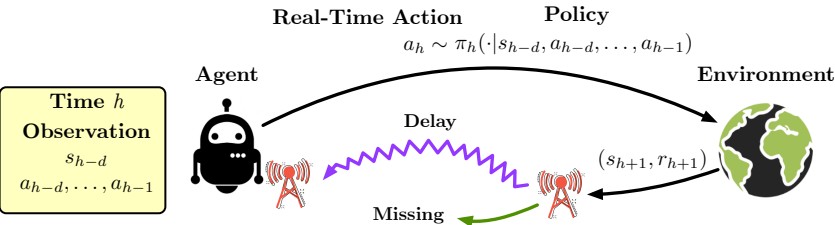

Figure 1: Reinforcement learning with impaired observability. At time $h$, the agent only observes the past state $s_{h-d}$ and actions $a_{h-d}, \ldots, a_{h-1}$. The policy depends on the observed information.

[2022], Chen et al. [2022], Liu et al. [2022], Zhong et al. [2022], Chen et al. [2023], where the unseen instant state can be inferred from recent observations. Unfortunately, MDPs with impaired observability do not fall into these benign subclasses. The reason behind this is that at each time step, a new observation, if any, is in fact a past state. Viewing it as an emission state of the current one leads to a time reversal posterior distribution depending on the underlying transitions, which suffers from the curse of history and makes the POMDP intractable. The problem becomes even harder if some observations get missing.

Empirical evidences suggested that efficient RL is possible even with impaired state observability Lizotte et al. [2008], Liu et al. [2014], Agarwal and Aggarwal [2021]. However, theoretical understanding of this problem is very limited. One notable work Walsh et al. [2007] studied learning with constant-time delayed observations. They identified subclasses of MDPs with nearly deterministic transitions that can be efficiently learned. Beyond this special case, efficient RL with impaired observability in MDPs with fully generality remains largely open.

Some recent works studied delayed feedback in MDPs Yang et al. [2023], Howson et al. [2023]. It is a fundamentally different problem where the agent's policy can still access real-time states but learning uses delayed data. Our problem is fundamentally harder because the agent's policy can only access the lossy and delayed history. See Section 1 for more discussions.

**Our results.** In this paper, we provide algorithms and regret analysis for learning the optimal policy in tabular MDPs with impaired observability. Note that this optimal policy is a different one from the optimal policy with full observability. To approach this problem, we construct an augmented MDP reformulation where the original state space is expanded to include available observations of past state and an action sequence. However, the expanded state space is much larger than the original one and naïve application of known methods would lead to exponentially large regret bounds. In our analysis, we exploit structure of the augmented transition model to achieve efficient learning and sharp regret bounds. The main results are summarized as follows.

● For MDPs with stochastic delays, we prove a sharp $\widetilde{O}(H^4\sqrt{SAK})$ regret bound (Theorem 4.1) comparing to the best feasible policy, Here $S$ and $A$ are the sizes of the original state and action spaces, respectively, $H$ is the horizon, and $K$ is the number of episodes. Here we allows the delay to be stochastic and conditionally independent given on current state and action. Moreover, we quantify the performance degradation of optimal value due to impaired observability, compared to optimal value of fully observable MDPs (Proposition B.2). We also showcase in Proposition 4.2 that a short delay does not reduce the optimal value, but slightly longer delay leads to substantial degradation.

● For MDPs with randomly missing observations, we provide an optimistic RL method that provably achieves $\widetilde{O}(\sqrt{H^3S^2AK})$ regret (Proposition 5.1). We also provide a sharper $\widetilde{O}(H^4\sqrt{SAK})$ regret in the case when the missing rate is sufficiently small (Theorem 5.2).

To our best knowledge, these results present a first set of theories for RL with delayed and missing observations. Remarkably, our regret bounds nearly match the minimax-optimal regret of standard MDP in their dependence on $S, A$ (noting that the target optimal policies are different in the two cases). It implies that RL with impaired observability are provably as efficient as RL with full observability (up to poly factors of $H$).

**Related work.** Efficient algorithms for learning in the standard setting of tabular MDPs without impaired observability has been extensively studied Kearns and Singh [2002], Brafman and Tennenholtz [2002], Jaksch et al. [2010], Dann and Brunskill [2015], Azar et al. [2017], Agrawal and Jia [2017],

Jin et al. [2018], Dann et al. [2019], Zanette and Brunskill [2019], Zhang et al. [2020], Domingues et al. [2021], where the minimax optimal regret is $\widetilde{\mathcal{O}}(\sqrt{H^3 SAK})$ Azar et al. [2017], Domingues et al. [2021].

The delayed observation studied in this paper is related to delayed feedback in Howson et al. [2023], Yang et al. [2023], yet the setup is fundamentally different. In delayed feedback, an agent sends a policy to the environment for execution. The environment executes the policy on behalf of the agent for an episode, but the whole trajectory will be returned to the agent after some episodes. The policy executed by the environment is able to "see" instant state and reward. It is Markov and not played by the agent. Our setting concerns learning executable policies when delayed or missing states appear within an episode. The policy is no longer Markov and can only prescribe action based on history. Therefore, the algorithms and analyses for delayed feedback MDPs are not applicable to our settings.

Despite the distinct settings, there are existing fruitful results in efficiently learning MDPs or bandits with delayed feedback. Stochastic delayed feedback in bandits is studied in Agarwal and Duchi [2011], Dudik et al. [2011], Joulani et al. [2013], Vernade et al. [2017, 2020], Gael et al. [2020], Lancewicki et al. [2021]. In the more challenging setting of reinforcement learning, Howson et al. [2023] considers tabular MDPs and Yang et al. [2023] generalizes to MDPs with function approximation and multi-agent settings.

On the other hand, results analyzing MDPs with missing observations are limited in literature, although missing data is a commonly recognized issue in applications García-Laencina et al. [2010], Jerez et al. [2010], Little et al. [2012], Emmanuel et al. [2021]. One notable result is Bouneffouf et al. [2020] for bandits with missing rewards.

**Notation**: For real numbers $a, b$, we denote $a \wedge b = \min\{a, b\}$. In episodic MDPs, we use the superscript $k$ to denote the index of episodes, and the subscript $h$ to denote the index of time. We denote $\mathbf{a}_{i:j} = \{a_i, \ldots, a_j\}$ as the collection of actions from time $i$ to $j$. For two probability distributions $\mu$ and $\nu$, we denote their total variation distance as $\|\mu - \nu\|_{\mathrm{TV}}$.

**MDP preliminary**: An episodic MDP is described by a tuple $(\mathcal{S}, \mathcal{A}, H, R, P)$, where $\mathcal{S}, \mathcal{A}$ are state and action spaces, respectively, $H$ is the horizon, $R = \{r_h\}_{h=1}^H$ is the reward function and $P = \{p_h\}_{h=1}^H$ is the transition probability. We primarily focus on tabular MDPs, where $S = |\mathcal{S}|$ and $A = |\mathcal{A}|$ are both finite. We also assume that the reward is uniformly bounded with $\|r_h\|_\infty \leq 1$ for any $h$. An agent will interact with the environment for $K$ episodes, hoping to find a good policy to maximize the cumulative reward. Within an episode, at the $h$-th step, the agent chooses an action based on the available information of the environment. After taking the action, the underlying environment produces a reward and transits to the next state. With full state observation, a policy $\pi$ maps instant state $s$ to an action $a$ or an action distribution. Given such a policy $\pi$, the value function is $V_h^\pi(s_1) = \mathbb{E}^\pi \left[ \sum_{h'=h}^H r_h(s_{h'}, a_{h'}) \big| s_h \right]$, where $\mathbb{E}^\pi$ is the policy induced expectation.

## 2 Problem formulation

In this work, we study MDPs with impaired observability. We focus on two practical settings: 1) delayed observations and 2) missing observations.

### 2.1 MDP with delayed observations

In any episode, we denote $d_h \in \{0, 1, \ldots\}$ as the observational delay of the state and reward at step $h$. That is, we receive $s_h$ and $r_h$ at time $h + d_h$. The delay time $d_h$ can be dependent on the state $s_h$ and action $a_h$ at time $h$. To facilitate analysis, we denote the inter-arrival time between the arrival of observations for step $h$ and $h + 1$ as $\Delta_h = d_{h+1} - d_h$. With delays, at time $h$, the nearest observable state is denoted as $s_{t_h}$, where $t_h = \mathrm{argmax}\left\{I : \sum_{i=0}^I \Delta_i \leq h\right\}$. Then the executable policy class

$$\Pi_{\mathrm{exec}} = \{\pi_h(\cdot | s_{t_h}, \mathbf{a}_{t_h:h-1}) \text{ for } h = 1, \ldots, H\}$$

chooses actions depending on the nearest visible state and history actions. We impose the following assumption on the interarrival time.

**Assumption 2.1** . The interarrival time $\Delta_h$ takes value in $\{0, 1, \ldots\}$. The distribution $\mathcal{D}_h(s_h, a_h)$ of $\Delta_h$ can depend on $(s_h, a_h)$, but is conditionally independent of the MDP transitions given $(s_h, a_h)$.

Assumption 2.1 does not impose any specific distributional assumption on $\Delta_h$, but only requires that the delayed observations arrive in order and at each time step, there is at most one new visible state and reward pair ($\Delta_h \geq 0$). A widely studied example of delays in literature is that the inter-arrival time is geometrically distributed Winsten [1959]. Then the observation sequence $\{h + d_h\}$ is known as a Bernoulli process, which is understood as the discretized version of a Poisson process.

Our delayed observation setting is newly proposed and substantially generalizes the Constant Delayed MDPs (CDMDPs) studied in Brooks and Leondes [1972], Bander and White III [1999], Katsikopoulos and Engelbrecht [2003], Walsh et al. [2007]. When $\Delta_h = 0$ being deterministic for all $h \geq 1$ and $k$, our observation delay coincides with CDMDPs. In CDMDPs, a new past observation is guaranteed to arrive at each time step. However, our delayed model can result in no new observation at some time steps.

Observation delay leads to difficulty in planning, as the agent can only infer the current state and then choose an action. Therefore, the policy is naturally history dependent. We summarize the interaction protocol of the agent with the environment in Protocol 1. At the end of each episode, we can collect

---

**Protocol 1** Interaction between the agent and the environment with delayed observations

1: **for** episode $k = 1, \ldots, K$ **do**
2:    **for** time $h = 1, \ldots, H$ **do**
3:       The agent observes a pair of new, if any, state and reward $(s_{t_h}^k, a_{t_h}^k)$. By memory, the agent also has access to past actions $\mathbf{a}_{t_h:h-1}^k$.
4:       The agent plays action $a_h^k$ according to some executable policy $\pi_h^k \in \Pi_{\text{exec}}$.
5:       The environment transits to next state $s_{h+1}^k \sim p_h(\cdot|s_h^k, a_h^k)$, which is unobservable to the agent. The environment also decides the delay at step $h + 1$ as $d_{h+1}^k = d_h^k + \Delta_h^k$ and $t_{h+1}^k$.
6:    **end for**
7:    The environment sends all unobserved pairs of state and reward as well as their corresponding delay time to the agent.
8: **end for**

---

all delayed observations, however, these observations are not used in planning. In reality, the agent can collect these observations by waiting after time $H$. Protocol 1 is similar to hindsight observability in POMDPs studied in Lee et al. [2023]. Yet their analysis for POMDPs is not directly transferable to our settings as mentioned in the introduction.

## 2.2 MDP with missing observations

In addition to the stochastic delay in observations, we also consider randomly missing observations. In applications, an agent interacts with the environment through some communication channel. The communication channel is often imperfect and thus, observation can be lost during transmission. This type of missing is permanent and we describe in the following assumption.

**Assumption 2.2** . Any pair of observation (state and reward) is independently observable in the communication channel. The observation rate is $\lambda_h$ depending on $h$, but independent of the MDP transitions. Moreover, there exists a constant $\lambda_0$ such that $\lambda_h \geq \lambda_0$ for any $h$. The agent will be informed when an observation is missing.

Equivalently, the missing observation rate in Assumption 2.2 is $1 - \lambda_h$ and assumes the upper bound of $1 - \lambda_0$. We will show later that this missing rate directly influences the learning efficiency in Section 5.

## 3 Construction of augmented MDPs

To tackle the limited observability, we expand the original state space and define an augmented MDP. It will serve as the basis for our subsequent theoretical analysis.

## 3.1 Augmented MDPs with expected reward

In the remainder of this section, we focus on the delayed observation case and defer the missing case to Section 5. Define $\tau_h = \{s_{t_h}, \mathbf{a}_{t_h:h-1}, \delta_{t_h}\}$ as the augmented state, where $\delta_{t_h} \in [0, \Delta_{t_h}]$ is the delayed steps after observing $(s_{t_h}, r_{t_h})$. Let $\mathcal{S}_{\text{aug}}$ denote the augmented state space of all possible $\tau$'s. Then the original MDP with delayed observations can be reformulated into a state-augmented one $\text{MDP}_{\text{aug}} = (\mathcal{S}_{\text{aug}}, \mathcal{A}, H, R_{\text{aug}}, P_{\text{aug}})$. The reward is defined as

$$r_{h,\text{aug}}(\tau_h, a_h) = \mathbb{E}\left[r_h(s_h, a_h)|\tau_h, a_h\right],$$

which is the expected reward given the nearest past state $s_{t_h}$ and history actions $\mathbf{a}_{t_h:h}$. We can define belief distribution $\mathfrak{b}_h(s|\tau_h) = \mathbb{P}(s_h = s|\tau_h)$. Then $r_{h,\text{aug}}(\tau_h, a_h) = \mathbb{E}_{s \sim \mathfrak{b}_h(\cdot|\tau_h)}[r(s, a_h)]$. Belief distributions are widely adopted in partially observed MDPs Ross et al. [2007], Poupart and Vlassis [2008]. We will frequently use the belief distribution to study the expressivity of $\Pi_{\text{exec}}$ in Section 4.2.

The transition probabilities $P_{\text{aug}}$ are sparse. For any $\tau_h = \{s_{t_h}, \mathbf{a}_{t_h:h-1}, \delta_{t_h}\}$ and $\tau_{h+1} = \{s_{t_{h+1}}, \mathbf{a}_{t_{h+1}:h}, \delta_{t_{h+1}}\}$, we have

| $p_{h,\text{aug}}(\tau_{h+1}|\tau_h, a_h)$ | Condition |
|---|---|
| $\mathtt{M}_a(\tau_h, \tau_{h+1})\theta_{\text{delay}}(s_{t_h}, a_{t_h}, \delta_{t_h})p_{t_h}(s_{t_{h+1}}|s_{t_h}, a_{t_h})$ | if $\delta_{t_{h+1}} = 0$ and $t_{h+1} = t_h + 1$ |
| $\mathtt{M}_a(\tau_h, \tau_{h+1})(1 - \theta_{\text{delay}}(s_{t_h}, a_{t_h}, \delta_{t_h}))$ | if $\delta_{t_{h+1}} = \delta_{t_h} + 1$ and $t_{h+1} = t_h$ |
| $0$ | otherwise |

where $\mathtt{M}_a(\tau_h, \tau_{h+1})$ indicates whether the rolling actions are matched, i.e.,

$$\mathtt{M}_a(\tau_h, \tau_{h+1}) = \mathbb{1}\{\mathbf{a}_{t_h:h-1} = \mathbf{a}_{t_{h+1}:h-1}\},$$

and $\theta_{\text{delay}}(s_{t_h}, a_{t_h}, \delta_{t_h})$ is defined as

$$\theta_{\text{delay}}(s_{t_h}, a_{t_h}, \delta_{t_h}) = \mathbb{P}(\Delta_{t_h} = \delta_{t_h}|s_{t_h}, a_{t_h}, \delta_{t_h}) = \frac{\mathbb{P}(\Delta_{t_h} = \delta_{t_h}|s_{t_h}, a_{t_h})}{1 - \sum_{\delta < \delta_{t_h}} \mathbb{P}(\Delta_{t_h} = \delta|s_{t_h}, a_{t_h})}.$$

The factored form of $\theta_{\text{delay}}(s_{t_h}, a_{t_h}, \delta_{t_h})p_{t_h}(s_{t_{h+1}}|s_{t_h}, a_{t_h})$ follows from the conditional independence in Assumption 2.1. We define $Q$-functions and value functions as follows. For any $\tau_h, a_h$ and policy $\pi \in \Pi_{\text{exec}}$, we have

$$Q_{h,\text{aug}}^{\pi}(\tau_h, a_h) = \mathbb{E}^{\pi}\left[\sum_{h'=h}^{H} r_{h,\text{aug}}(\tau_{h'}, a_{h'})\Big|\tau_h, a_h\right] \quad \text{and}$$

$$V_{h,\text{aug}}^{\pi}(\tau_h) = \left\langle Q_{h,\text{aug}}^{\pi}(\tau_h, \cdot), \pi_h(\cdot|\tau_h)\right\rangle.$$

We note that $V_h^{\pi}$ is equivalent to $V_{h,\text{aug}}^{\pi}$ for the same executable policy $\pi \in \Pi_{\text{exec}}$. We also denote $\mathcal{P}_{h,\text{aug}}$ as the transition operator corresponding to $P_{\text{aug}}$. It can be checked that

$$Q_{h,\text{aug}}^{\pi}(\tau_h, a_h) = r_{h,\text{aug}}(\tau_h, a_h) + [\mathcal{P}_{h,\text{aug}}V_{h,\text{aug}}^{\pi}](\tau_h, a_h).$$

$\text{MDP}_{\text{aug}}$ also appears in makes all the policies in $\Pi_{\text{exec}}$ executable and Markov. Meanwhile, the reward function keeps track of all the expected reward for $H$ steps. Although the expanded state space $\mathcal{S}_{\text{aug}}$ is much more complicated than the original state space $\mathcal{S}$, the sparse structures in the transition probabilities still allow an efficient exploration. We note that $p_{h,\text{aug}}$ only depends on the delay distribution and one-step Markov transitions. However, there is still one caveat for learning in $\text{MDP}_{\text{aug}}$ – the reward function depends belief distributions, which involve multi-step transitions.

## 3.2 Augmented MDPs with past reward

To tackle the aforementioned challenge, we further define $\widetilde{\text{MDP}}_{\text{aug}} = (\widetilde{\mathcal{S}}_{\text{aug}}, \mathcal{A}, \widetilde{H}, \widetilde{R}_{\text{aug}}, \widetilde{P}_{\text{aug}})$ that shares the optimal policy in $\text{MDP}_{\text{aug}}$ with an enlonged horizon $\widetilde{H} = 2H$. The state space $\widetilde{\mathcal{S}}_{\text{aug}}$ consists of any $\tau_h = \{s_{t_h}, \mathbf{a}_{t_h:h \wedge H}, \delta_{t_h}\}$. Comparing to $\mathcal{S}_{\text{aug}}$, we cut off the action at horizon $H$, since $a_h$ for $h > H$ has no influence on the state and reward in time $[1, H]$. The reward function is defined as

$$\widetilde{r}_{h,\text{aug}}(\tau_h, a_h) = r_{t_h}(s_{t_h}, a_{t_h})\mathbb{1}\{\delta_{t_h} = 0\}\mathbb{1}\{t_h \in \{1, \dots, H\}\}.$$

By definition, $\widetilde{r}_{\mathrm{aug}}(\tau_h, a_h)$ is a past reward. More importantly, $\widetilde{r}_{h,\mathrm{aug}}(\tau_h, a_h)$ zeros out rewards outside the original horizon $H$. Meanwhile, between the arrival of two consecutive state observations, the reward only counts once. Lastly, the transition probabilities are

| $\widetilde{p}_{h,\mathrm{aug}}(\tau_{h+1}\mid\tau_h, a_h)$ | Condition |
|---|---|
| $\mathrm{M}_a(\tau_h, \tau_{h+1})\theta_{\mathrm{delay}}(s_{t_h}, a_{t_h}, \delta_{t_h})p_{t_h}(s_{t_{h+1}}\mid s_{t_h}, a_{t_h})$ | if $\delta_{t_{h+1}} = 0, t_{h+1} = t_h + 1$ and $h < H$ |
| $\mathrm{M}_a(\tau_h, \tau_{h+1})(1 - \theta_{\mathrm{delay}}(s_{t_h}, a_{t_h}, \delta_{t_h}))$ | if $\delta_{t_{h+1}} = \delta_{t_h} + 1, t_{h+1} = t_h$ and $h < H$ |
| $\mathrm{M}_a(\tau_h, \tau_{h+1})p_{t_h}(s_{t_h+1}\mid s_{t_h}, a_{t_h})$ | if $\delta_{t_{h+1}} = 0, t_{h+1} = t_h + 1$ and $h > H$ |
| $0$ | otherwise |

We interpret the transitions as follows. When $h \leq H$, the transition is the same as $\mathrm{MDP}_{\mathrm{aug}}$. When $h > H$, we simply wait for unobserved states and rewards to come. As mentioned, actions taken beyond time $H$ are irrelevant. We build an equivalence in the expected values of $\mathrm{MDP}_{\mathrm{aug}}$ and $\widetilde{\mathrm{MDP}}_{\mathrm{aug}}$.

**Proposition 3.1.** Let $\mathrm{MDP}_{\mathrm{aug}}$ and $\widetilde{\mathrm{MDP}}_{\mathrm{aug}}$ be defined as in the previous paragraphs. Then for any initial state $\tau_1$ and any policy $\pi = \{\pi_h\}_{h=1}^{H} \in \Pi_{\mathrm{exec}}$, it holds that

$$\mathbb{E}^\pi\left[\sum_{h=1}^{H} r_{h,\mathrm{aug}}(\tau_h, a_h)\Big|\tau_1\right] = \mathbb{E}^\pi\left[\sum_{h=1}^{\widetilde{H}} \widetilde{r}_{h,\mathrm{aug}}(\tau_h, a_h)\Big|\tau_1\right],$$

where in the right-hand side, the policy for steps $H + 1$ to $\widetilde{H}$ is arbitrary.

The proof is provided in Appendix A.1. Proposition 3.1 implies that learning in $\mathrm{MDP}_{\mathrm{aug}}$ until time $H$ is equivalent to that in $\widetilde{\mathrm{MDP}}_{\mathrm{aug}}$ for $\widetilde{H}$ steps.

# 4 RL with delayed observations and regret bound

In this section, we provide regret analysis of learning in MDPs with stochastic delays. For the sake of simplicity, we assume the reward is known, however, extension to unknown reward causes no real difficulty. Motivated by the augmented MDP reformulation, we introduce our learning algorithm in Algorithm 2. In Line 5, unobserved states and rewards are returned to the agent as described in Protocol 1. Using the data set, we construct bonus functions compensating the uncertainty in *one-step* transitions of the original MDP. This largely sharpens the confidence region, yet still ensures a valid optimism. We emphasize that in Line 9, we are planning on $\widetilde{\mathrm{MDP}}_{\mathrm{aug}}$ involving the augmented transitions and expanded states of $\tau \in \widetilde{\mathcal{S}}_{\mathrm{aug}}$. Only in this way, we can obtain an executable policy in delayed MDPs. The planning complexity is $SA^H$ though.

## 4.1 Regret bound

We define regret in delayed MDP as

$$\mathtt{Regret}(K) = \sum_{k=1}^{K} \max_{\pi\in\Pi_{\mathrm{exec}}} V_1^\pi(s_1^k) - \sum_{k=1}^{K} V_1^{\pi_k}(s_1^k),$$

where $V_1^\pi$ is the value function of the original MDP. Although the regret here is defined on the original MDP, it is equivalent to the regret of the same policy on $\mathrm{MDP}_{\mathrm{aug}}$ and further $\widetilde{\mathrm{MDP}}_{\mathrm{aug}}$ by Proposition 3.1. Note that we are comparing with the best executable policy. The performance degradation caused by observation delay is discussed in Section 4.2. The following theorem bounds the regret.

**Theorem 4.1** (Regret bound for Delayed MDP). Suppose Assumption 2.1 holds. Let $\gamma \in (0, 1)$ be any failure probability. With probaiblity $1 - \gamma$, the regret of Algorithm 2 satisfies

$$\mathtt{Regret}(K) \leq c\left(H^4\sqrt{SAK\iota} + H^4 S^2 A \iota^2\right),$$

where $\iota = \log\frac{SAHK}{\gamma}$ and $c$ is a constant.

The proof is provided in Appendix B.1. We discuss several implications.

---

**Algorithm 2** Policy learning for delayed MDPs using $\widetilde{\text{MDP}}_{\text{aug}}$

---

1: **Input**: Original horizon $H$, extended horizon $\widetilde{H}$, policy class $\Pi_{\text{exec}}$, failure probability $\gamma$.
2: **Init**: $V_{\widetilde{H}+1}(\tau) = 0$ and $Q_{\widetilde{H}}(\tau, a) = H$ for any $\tau$ and $a$, data set $\mathcal{D}^0 = \emptyset$, initial policy $\pi^0$.
3: **for** episode $k = 1, \ldots, K$ **do**
4:    Execute policy $\pi^{k-1}$ for $\widetilde{H}$ steps.
5:    After the episode ends, collect data $\mathcal{D}^k = \mathcal{D}^{k-1} \cup \{(s_h^k, a_h^k, r_h^k, \Delta_h^k)\}_{h=1}^H$.
6:    On data set $\mathcal{D}^k$, compute counting numbers $N_h^k(s_h, a_h)$, $N_h^k(s_h, a_h, s_{h+1})$ and
    $N_h^k(s_h, a_h, \delta_h) = \sum_{j=1}^k \mathbb{1}\{s_h^j = s_h, a_h^j = a_h, \Delta_h^j = \delta_h\}$.
7:    Estimate transition probabilities and delay distributions via

$$\widehat{p}_h^k(s_{h+1}|s_h, a_h) = \frac{N_h^k(s_h, a_h, s_{h+1})}{N_h^k(s_h, a_h)}, \quad \text{and} \quad \widehat{\theta}_{\text{delay}}^k(s_h, a_h, \delta_h) = \frac{N_h^k(s_h, a_h, \delta_h)}{\sum_{\delta \geq \delta_h} N_h^k(s_h, a_h, \delta)}.$$

   Then estimators of $\widetilde{p}_{h,\text{aug}}$ in $\widetilde{\text{MDP}}_{\text{aug}}$ is computed using $\widehat{p}_h^k$ and $\widehat{\theta}_{\text{delay}}^k$.
8:    Set bonus function as

$$b_h^k(\tau_h, a_h) = cH \left( \sqrt{\frac{(H \wedge D)\iota}{N_{t_h}^k(s_{t_h}, a_{t_h}, \delta_{t_h})}} + \sqrt{\frac{(H \wedge D)\iota}{N_{t_h}^k(s_{t_h}, a_{t_h})}} \right)$$

   for $\iota = \log \frac{SAKH}{\gamma}$ and $c$ sufficiently large.
9:    Run optimistic value iteration in $\widetilde{\text{MDP}}_{\text{aug}}$ for $\widetilde{H}$ steps and obtain $\pi^k \in \Pi_{\text{exec}}$.
10: **end for**
11: **Return**: Learned policy $\pi_{1:H}^k$ for $k = 1, \ldots, K$.

---

**Sharp dependence on $S$ and $A$**    Theorem 4.1 has a sharp dependence on $S$ and $A$, although the expanded state space $\widetilde{\mathcal{S}}_{\text{aug}}$ has a cardinality bounded by $SA^H$. Naïvely learning and planning in $\widetilde{\text{MDP}}_{\text{aug}}$ would suffer from the exponential enlargement of $A^H$. However, we identify the sparse structures in the transition probabilities. As can be seen, $\widetilde{p}_{h,\text{aug}}$ only involves one-step transitions in the original MDP and some conditionally independent delay distributions. Such structures lead to a rather easy estimation of $\widetilde{p}_{h,\text{aug}}$, which can be constructed from the estimators of one-step transitions in the original MDP. Meanwhile, the sparse structures make exploration in $\widetilde{\text{MDP}}_{\text{aug}}$ efficient, due to many unreachable states.

**Effect of the delay distribution and delay length**    Theorem 4.1 holds for arbitrary conditionally independent delay distributions, even include heavy-tailed distributions. In the worst case of unbounded delays, Theorem 4.1 gives rise to a $\mathcal{O}(H^4\sqrt{SAK\iota})$ regret. The reason to this is that if the delay is larger than $H$, then the corresponding state will only be observed after an episode ends and won't be used in planning. Therefore, we can truncate the delay at $H$, regardless of its tail distributions.

When the maximal length of delay is bounded by $D < H$, e.g., CDMDPs with $d_h = D$ for any $h$, Theorem 4.1 implies that the regret is bounded by

$$\texttt{Regret}(K) \leq c \left( (D+1)^{5/2}\sqrt{H^3 SAK\iota} + H^4 S^2 A\iota^2 \right)$$

for a constant $c$. A proof is provided in Appendix B.2. As can be seen, as the length of delay increases, the regret bound enlarges, reflecting the increased difficulty of long delays. Moreover, when $D = 0$, that is, no observation delays, the regret bound recovers that in standard MDPs.

### 4.2    Performance degradation of policy class $\Pi_{\text{exec}}$

This section devotes to quantify the performance degradation caused by delayed observations. In particular, we bound the value difference between the best executable policy and the best Markov policy in a no delay environment. Recall that $V_1$ is the value function of the original MDP. We denote

$$\pi_{\text{nodelay}}^* = \text{argmax}_\pi V_1^\pi(s_1) \quad \text{and} \quad \pi_{\text{delay}}^* = \text{argmax}_{\pi \in \Pi_{\text{exec}}} V_1^\pi(s_1)$$

as the best vanilla optimal policy and executable policy, respectively. The values achieved by $\pi^*_{\text{nodelay}}$ and $\pi^*_{\text{delay}}$ are denoted as $V^*_{1,\text{nodelay}}(s_1)$ and $V^*_{1,\text{delay}}(s_1)$, respectively. The gap between $V^*_{1,\text{nodelay}}$ and $V^*_{1,\text{delay}}$ quantifies the performance degradation, which is denoted as $\text{gap}(s_1) = V^*_{1,\text{nodelay}}(s_1) - V^*_{1,\text{delay}}(s_1)$. We bound $\text{gap}$ in Proposition B.2 in Appendix due to space limit.

In a nutshell, we show that the performance degradation $\text{gap}$ is highly relevant to the belief distribution $\mathfrak{b}_h(\cdot|\tau)$. When $\mathfrak{b}_h(\cdot|\tau)$ is evenly spread, meaning that the entropy of $\mathfrak{b}_h$ is high and inferring the current unseen state is difficult, we potentially suffer from a large $\text{gap}$. On the contrary, when $\mathfrak{b}_h(\cdot|\tau)$ is nearly deterministic, the performance degradation is small. In the special case of deterministic transitions, we have $\text{gap} = 0$.

### 4.3 The (mysterious) effect of delay on the optimal value

To further understand the effect of the delay on the optimal value, we provide the following dichotomy. On the one hand, we show that there exists an MDP instance, such that a constant delay of $d$ steps does not hurt the performance. On the other hand, in the same MDP instance, a constant delay of $d + 1$ steps suffers from a constant performance drop.

**Proposition 4.2.** Consider constant delayed MDPs. Fix a positive integer $d < H$. Then there exists an MDP instance such that the following two items hold simultaneously.

- When delay is $d$, it holds that $\frac{1}{K} \sum_{k=1}^{K} \text{gap}(s_1^k) = 0$.

- When delay is $d + 1$, it holds that $\frac{1}{K} \sum_{k=1}^{K} \text{gap}(s_1^k) \geq \frac{1}{2} - \sqrt{\frac{1}{2K} \log \frac{1}{\gamma}}$, with probability $1 - \gamma$.

The proof is provided in Appendix B.4. We remark that Proposition 4.2 says that observation delay can be dangerous, even with the slightest possible number of steps. The idea behind Proposition 4.2 is consistent with the analysis on $\text{gap}$. In particular, we construct an MDP instance demonstrated in Figure 2. The reward vanishes at all times but $d + 1$. When delay is $d$, the initial state $s_1$ is revealed and the policy can choose the best action to receive a reward. When delay is $d + 1$, however, there is always a $1/2$ probability of missing the best action for any policy, which leads to a constant performance degradation.

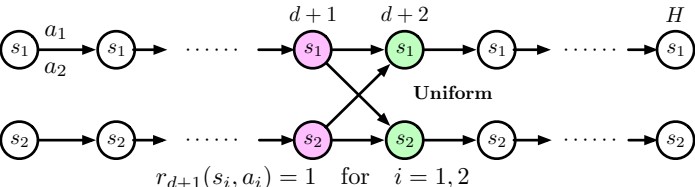

Figure 2: MDP instance on two states with two actions. The transition is lazy until time $d$. Then the transition is uniform regardless of actions for time $d + 1$. Reward is nonzero only at time $d + 1$. This is an example with a delay of length $d$ causes no degradation and a delay of $d + 1$ causes a constant performance degradation.

## 5 RL with missing observations and regret analysis

We now switch our study to MDPs with missing observations. In such an environment, executable policies share the same structures as delayed MDPs, where an action is taken based on available history information. Compared to delayed observations, learning with missing observations is more challenging. Since unobserved states and rewards are never revealed, we are suffering from information loss. Besides, we will frequently deal with multi-step transitions, due to missing observations between two consecutive visible states.

### 5.1 Optimistic planning with missing observations

Despite the difficulty, we present here algorithms that are efficient in learning and planning for MDPs with missing observations. We begin with an optimistic planning algorithm in Algorithm 3. To unify the notation, we denote $s_h^k = \emptyset$ and $r_h^k = \emptyset$ as missing the observation.

---
**Algorithm 3** Optimistic planning for MDPs with missing observations
---
1: **Input**: Horizon $H$, observable rate $\lambda_h$.
2: **Init**: $\mathcal{B}^0 = \Theta$ to be all possible tabular MDPs, data set $\mathcal{D}^0 = \emptyset$.
3: **for** episode $k = 1, \ldots, K$ **do**
4:      Set policy $\pi^k = \operatorname{argmax}_{\pi \in \Pi_{\mathrm{exec}}} \max_{\theta \in \mathcal{B}^k} V_{1,\theta}^\pi(s_1^k)$.
5:      Play policy $\pi^k$ and collect data $\mathcal{D}^{k-1} \cup \{(s_h^k, a_h^k, r_h^k)\}_{h=1}^H$.
6:      Compute counting number $N_h^k(s,a) = \sum_{j=1}^k \mathbb{1}\{s_h^j = s, a_h^j = a, s_{h+1}^j \neq \emptyset\}$.
7:      Update confidence set

$$\mathcal{B}^k = \left\{\theta : \|\widehat{p}_h^k(\cdot|s,a) - p_h^\theta(\cdot|s,a)\|_{\mathrm{TV}} \leq c\sqrt{\frac{S\iota}{N_h^k(s,a)}} \text{ for all } (h,s,a)\right\} \cap \mathcal{B}^{k-1},$$

     where $\widehat{p}_h^k(s'|s,a) = \frac{N_h^k(s,a,s')}{N_h^k(s,a)}$ and $c$ is some constant.
8: **end for**
---

The majority of the algorithm resembles the typical optimistic planning Jaksch et al. [2010] but with some notable differences. In Line 4, the value function $V_{1,\theta}$ is for the original MDP with transition probabilities parameterized by $\theta$. Different from the typical optimistic planning, the underlying MDP here obeys the stochastic observable model in Assumption 2.2. Therefore, the value $V_{1,\theta}$ is the sum of all possible values under missing observations. When counting $N_h^k(s,a)$ in Line 6, we exclude data tuples missing the next state, which inevitably slows down the learning curve. Nonetheless, the effect of missing only contributes as a scaling factor in the regret.

**Proposition 5.1.** Suppose Assumption 2.2 holds with $\lambda_h$ known. Given a failure probability $\gamma$, with probability $1 - \gamma$, the regret of Algorithm 4 satisfies

$$\mathtt{Regret}(K) \leq c\left(\left\lceil \frac{1}{-\log(1-\lambda_0^2)}\right\rceil \sqrt{H^3 S^2 A K \iota^3} + \sqrt{H^4 K \iota}\right),$$

where $\iota = \log\frac{SAHK}{\gamma}$ and $c$ is a constant.

The proof is provided in Appendix C.1. Proposition 5.1 is optimal in the $K$ dependence and achieves an $S^2A$ dependence on the complexity of the underlying MDP. In the extreme case of $\lambda_0 \approx 0$, which implies that every state and reward are hardly observable, we have $\mathtt{Regret}(K) = \widetilde{\mathcal{O}}\left(\frac{1}{\lambda_0^2}\sqrt{H^3 S^2 A K}\right)$. Here $\lambda_0^2$ is the probability of observing two consecutive states for estimating the transition probabilities. Proposition 5.1 requires knowledge of observable rate $\lambda_h$. This is not a restrictive condition, as estimating $\lambda_h$ from Bernoulli random variables is much easier than estimating transition probabilities.

### 5.2 Model-based planning using augmented MDPs

Proposition 5.1 has a lenient dependence on the missing rate $1 - \lambda_0^2$, nonetheless, is not sharp on the dependence of $S$. We next show that the augmented MDP approach is effective to tackle missing observations, when the observable rate satisfies additional conditions. Specifically, we assume that the observable rate $\lambda_h$ is independent of $(s,a)$. We utilize the $\mathtt{MDP}_{\mathrm{aug}}$ reformulation, except that we redefine the transition probabilities as

$$p_{h,\mathrm{aug}}(\tau_{h+1}|\tau_h, a_h) = \begin{cases} \lambda_h p_h(s_{h+1}|s_{t_h}, \mathbf{a}_{t_h:h}) & \text{if } t_{h+1} = h+1 \\ \mathtt{M}_a(\tau_{h+1}, \tau_h)(1 - \lambda_h) & \text{if } t_{h+1} = t_h \\ 0 & \text{otherwise} \end{cases}.$$

The first case in $p_{h,\mathrm{aug}}$ corresponds to receiving the state observation at time $h+1$. In contrast to the delayed MDPs, the transition probabilities here potentially rely on multi-step transitions in the original MDP. The second case of the transition corresponds to missing the observation. We summarize the policy learning procedure in Algorithm 4 in Appendix C.2, which is similar to Algorithm 2, but with a new bonus function. The following theorem shows that Algorithm 4 is asymptotically efficient when the observable rate is relatively high.

**Theorem 5.2.** Suppose Assumption 2.2 holds with $\lambda_0 \geq 1 - A^{-(1+v)}$ for some positive constant $v$. Given a failure probability $\gamma$, with probability $1 - \gamma$, the regret of Algorithm 4 satisfies

$$\texttt{Regret}(K) \leq c \left( H^4 \sqrt{SAK\iota^3} + S^2 \sqrt{H^9 K^{\frac{1}{(1+v)}} \iota^6} \right),$$

where $\iota = \log \frac{SAHK}{\gamma}$ and $c$ is a constant.

The proof is provided in Appendix C.2. Some remarks are in order.

$SA$ **rate when** $K$ **is large**   When the number of episodes $K \geq S^{3(1+v)/v}$, the first term $H^4 \sqrt{SAK\iota^3}$ in the regret bound dominates and attains a sharp dependence on $S$ and $A$. However, when the number of episodes are limited, the regret bound has a worse dependence on the state space size $S$. We also observe that as the missing rate $\lambda$ becomes small (equivalently, $v$ becomes large), the regret is close to $\widetilde{O}(H^4 \sqrt{SAK\iota^3})$.

**Observable rate smaller than** $1 - 1/A$   Theorem 5.2 holds for an observable rate $\lambda_0 > 1 - 1/A$. The intuition behind is that to fully explore all the actions when a state observation is missing takes $A$ trials. Therefore, in expectation, we will encounter a missing observation at least every $A$ episodes as long as $\lambda_0 > 1 - 1/A$. Nonetheless, when $\lambda_0 \leq 1 - 1/A$, the regret bound remains curiously underexplored. We conjecture that $\lambda_0 = 1 - 1/A$ is a critical point distinguishes unique strategies for learning and planning in MDPs with missing observations. A detailed analysis goes beyond the scope of the current paper.

**Proof sketch**   The proof of Theorem 5.2 adapts the analysis of model-based UCBVI algorithms Azar et al. [2017]. Let $m$ denote the maximal length of consecutive missing observations. We denote $\mathcal{E}_m$ as the event when the maximal length of consecutive missing is less than $m$. On event $\mathcal{E}_m$, a naïve analysis leads to a $\widetilde{\mathcal{O}} \left( \sqrt{\text{poly}(H)SA^{m+1}K} \right)$ regret, in observation to the size of the expanded state space $\mathcal{S}_{\text{aug}}$. However, our analysis circumvents the $A^m$ dependence by exploiting the occurrence of consecutive missing observations is rare (Lemma C.3). On the complement of event, the regret is bounded by $KH(1 - \mathbb{P}(\mathcal{E}_m))$. Summing up the two parts and choosing a proper $m$ yield our result.

# 6   Conclusion

In this paper, we have studied learning and planning in impaired observability MDPs. We focus on MDPs with delayed and missing observations. Specifically, for delayed observations, we have shown an efficient $\widetilde{O}(H^4 \sqrt{SAK})$ regret. For missing observations, we have provided an optimistic planning algorithm achieving an $\widetilde{O}(\sqrt{H^3 S^2 AK})$ regret. If the missing rate is relatively small, we have shown an efficient $\widetilde{O}(H^4 \sqrt{SAK})$ regret bound. Further, we have characterized the performance drop caused by impaired observability compared to full observability.

# Acknowledgement

The work of H. V. Poor was supported in part by a grant from the C3.ai Digital Transformation Institute. Mengdi Wang acknowledges the support by NSF grants DMS-1953686, IIS-2107304, CMMI-1653435, CPS-2312093, ONR grant 1006977, and C3.AI.

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

# A    Omitted proof in Section 3

## A.1    Proof of Proposition 3.1

*Proof.* Consider an arbitrary fixed inter-arrival pattern $\Delta_0, \Delta_1, \ldots, \Delta_{H-1}$. We show that the expected accumulated rewards under this inter-arrival pattern are identical for $\mathtt{MDP}_{\mathrm{aug}}$ and $\widetilde{\mathtt{MDP}}_{\mathrm{aug}}$. In $\widetilde{\mathtt{MDP}}_{\mathrm{aug}}$, we have

$$\mathbb{E}^\pi \left[ \sum_{h=1}^{\widetilde{H}} \widetilde{r}_{h,\mathrm{aug}}(\tau_h, a_h) \,\Big|\, \tau_1, \Delta_0, \ldots, \Delta_{H-1} \right]$$

$$\stackrel{(i)}{=} \mathbb{E}^\pi \left[ \sum_{h=1}^{\widetilde{H}} \widetilde{r}_{t_h,\mathrm{aug}}(s_{t_h}, a_{t_h}) \mathbb{1}\{\delta_{t_h} = 0\} \mathbb{1}\{t_h \in \{1, \ldots, H\}\} \,\Big|\, \tau_1, \Delta_0, \ldots, \Delta_{H-1} \right]$$

$$\stackrel{(ii)}{=} \mathbb{E}^\pi \left[ \sum_{h=1}^{H} r(s_h, a_h) \,\Big|\, \tau_1, \Delta_0, \ldots, \Delta_{H-1} \right]$$

$$= \mathbb{E}^\pi \left[ \sum_{h=1}^{H} r_{h,\mathrm{aug}}(\tau_h, a_h) \,\Big|\, \tau_1, \Delta_0, \ldots, \Delta_{H-1} \right],$$

where equality $(i)$ invokes the definition of $\widetilde{r}_{h,\mathrm{aug}}$ and equality $(ii)$ eliminates zero reward terms. Now taking expectation over all possible inter-arrival patterns, we deduce

$$\mathbb{E}^\pi \left[ \sum_{h=1}^{\widetilde{H}} \widetilde{r}_{\mathrm{aug}}(\tau_h, a_h) \,\Big|\, \tau_1 \right] = \mathbb{E}^\pi \left[ \sum_{h=1}^{H} r_{h,\mathrm{aug}}(s_h, a_h) \,\Big|\, \tau_1 \right].$$

The proof is complete. $\qquad\square$

# B    Omitted proofs in Section 4

## B.1    Proof of Theorem 4.1

*Proof.* We adapt the main steps from Azar et al. [2017] for proving the theorem. The proof consists of verifying a valid optimism and developing a regret analysis. We denote $\widetilde{Q}^*_{h,\mathrm{aug}}$ as the optimal $Q$-function for $\widetilde{\mathtt{MDP}}_{\mathrm{aug}}$. When analyzing the regret, we also denote $\widetilde{Q}^k_{h,\mathrm{aug}}$ as the optimal $Q$-function in the $k$-th episode.

**Valid optimism**    To begin with, we verify that the choice of the bonus functions leads to a valid optimism in the following lemma.

**Lemma B.1.** Given any failure probability $\gamma < 1$, we set a bonus as

$$b_h^k(\tau_h, a_h) = c_A H \left( \sqrt{\frac{H\iota}{N_{t_h}(s_{t_h}, a_{t_h}, \delta_{t_h})}} + \sqrt{\frac{H\iota}{N_{t_h}(s_{t_h}, a_{t_h})}} \right),$$

where $\iota = \log\left(\frac{SAHK}{\gamma}\right)$ and $c_A$ is a constant. Then with probability $1 - \gamma$, it holds

$$\widetilde{Q}^k_{h,\mathrm{aug}}(\tau_h, a_h) \geq \widetilde{Q}^*_{h,\mathrm{aug}}(\tau_h, a_h), \quad \widetilde{V}^k_{h,\mathrm{aug}}(\tau_h) \geq \widetilde{V}^*_{h,\mathrm{aug}}(\tau_h) \quad \text{for any} \quad (k, h, \tau_h, a_h).$$

*Proof of Lemma B.1.* We compute the cardinality of the expanded state space $\widetilde{\mathcal{S}}_{\mathrm{aug}}$ as

$$|\widetilde{\mathcal{S}}_{\mathrm{aug}}| \stackrel{(i)}{=} \sum_{i=0}^{H} HSA^i = HS \frac{A^{H+1} - 1}{A - 1} \leq 2HSA^H.$$

For a fixed episode $k$, we show by backward induction that the assertion in Lemma B.1 holds. To ease the presentation, we omit all superscripts $k$, all subscripts "aug", as well as the tilde $\widetilde{\cdot}$ notation.

When $h = \widetilde{H} + 1$, the base assertion holds immediately. Suppose the assertion is true for time $h + 1$. At time $h$, for any fixed $(\tau_h, a_h)$, if $Q_h(\tau_h, a_h) = H$, the assertion holds true. Otherwise, we have

$$Q_h(\tau_h, a_h) - Q_h^*(\tau_h, a_h) = [\widehat{\mathcal{P}}_h V_{h+1}](\tau_h, a_h) - [\mathcal{P}_h V_{h+1}^*](\tau_h, a_h) + b_h^k(\tau_h, a_h)$$

$$\geq \underbrace{\left([\widehat{\mathcal{P}}_h - \mathcal{P}_h] V_{h+1}^*\right)(\tau_h, a_h)}_{(A)} + b_h^k(\tau_h, a_h).$$

We show a lower bound on $(A)$. If $h \geq H$, expanding the transition kernel $\mathcal{P}_h$ leads to

$$(A) = \sum_{\tau_{h+1}} V_{h+1}^*(\tau_{h+1})(\widehat{p}_h(\tau_{h+1}|\tau_h, a_h) - p_h(\tau_{h+1}|\tau_h, a_h))$$

$$\overset{(i)}{=} \sum_{s_{t_h+1}} V_{h+1}^*(\tau_{h+1})(\widehat{p}_{t_h}(s_{t_h+1}|s_{t_h}, a_{t_h}) - p_{t_h}(s_{t_h+1}|s_{t_h}, a_{t_h}))$$

$$\overset{(ii)}{\geq} -c_{A,1} H \sqrt{\frac{H\iota}{N_{t_h}(s_{t_h}, a_{t_h})}},$$

where equality $(i)$ requires $\tau_{h+1}$ to take $s_{t_h+1}$ as the new state observation, and inequality $(ii)$ follows from the Hoeffding's inequality (Lemma D.2) with a constant $c_{A,1}$. Note that the $H\iota$ term in the numerator comes from a union bound over $\widetilde{\mathcal{S}}_{\mathrm{aug}} \times \mathcal{A}$.

On the other hand, if $h < H$, expanding the transition kernel $\mathcal{P}_h$ yields

$$(A) = \sum_{\tau_{h+1}} V_{h+1}^*(\tau_{h+1})\left(\widehat{p}_h(\tau_{h+1}|\tau_h, a_h) - p_h(\tau_{h+1}|\tau_h, a_h)\right)$$

$$= \underbrace{\sum_{\tau_{h+1}} V_{h+1}^*(\tau_{h+1})\left(\widehat{p}_h(\tau_{h+1}|\tau_h, a_h) - p_h(\tau_{h+1}|\tau_h, a_h)\right)\mathbb{1}\{\delta_{t_{h+1}} = 0\}\mathbb{1}\{t_{h+1} = t_h + 1\}}_{(A_1)}$$

$$+ \underbrace{\sum_{\tau_{h+1}} V_{h+1}^*(\tau_{h+1})\left(\widehat{p}_h(\tau_{h+1}|\tau_h, a_h) - p_h(\tau_{h+1}|\tau_h, a_h)\right)\mathbb{1}\{\delta_{t_{h+1}} = \delta_{t_h} + 1\}\mathbb{1}\{t_{h+1} = t_h\}}_{(A_2)}.$$

Note that $(A_1)$ accounts for receiving a new state observation in $\tau_{h+1}$, and $(A_2)$ accounts for no new state observation. We tackle these two terms separately. For $(A_1)$, we have

$$(A_1)$$
$$= \sum_{s_{t_h+1}} V_{h+1}^*(\tau_{h+1})\left((1 - \widehat{\theta}_{t_h}(s_{t_h}, a_{t_h}, \delta_{t_h}))\widehat{p}_{t_h}(s_{t_h+1}|s_{t_h}, a_{t_h}) - (1 - \theta_{t_h}(s_{t_h}, a_{t_h}, \delta_{t_h}))p_{t_h}(s_{t_h+1}|s_{t_h}, a_{t_h})\right)$$

$$= \sum_{s_{t_h+1}} V_{h+1}^*(\tau_{h+1})\left(\left(1 - \widehat{\theta}_{t_h}(s_{t_h}, a_{t_h}, \delta_{t_h})\right) - (1 - \theta_{t_h}(s_{t_h}, a_{t_h}, \delta_{t_h}))\right)\widehat{p}_{t_h}(s_{t_h+1}|s_{t_h}, a_{t_h})$$

$$+ \sum_{s_{t_h+1}} V_{h+1}^*(\tau_{h+1})(1 - \theta_{t_h}(s_{t_h}, a_{t_h}, \delta_{t_h}))\left(\widehat{p}_{t_h}(s_{t_h+1}|s_{t_h}, a_{t_h}) - p_{t_h}(s_{t_h+1}|s_{t_h}, a_{t_h})\right)$$

$$\overset{(i)}{\geq} -H\left|\widehat{\theta}_{t_h}(s_{t_h}, a_{t_h}, \delta_{t_h}) - \theta_{t_h}(s_{t_h}, a_{t_h}, \delta_{t_h})\right| - c_{A,2} H \sqrt{\frac{H\iota}{N_{t_h}(s_{t_h}, a_{t_h})}},$$

where in $(i)$, the first term is the estimation error of $\widehat{\theta}$ using the collected data, the second term follows from Hoeffding's inequality, and $c_{A,2}$ is an absolute constant. For $(A_2)$, we have

$$(A_2) \geq -H\left|\widehat{\theta}_{t_h}(s_{t_h}, a_{t_h}, \delta_{t_h}) - \theta_{t_h}(s_{t_h}, a_{t_h}, \delta_{t_h})\right|,$$

since $\tau_{h+1}$ is now uniquely determined. Summing up $(A_1)$ and $(A_2)$, we obtain

$$(A) = (A_1) + (A_2) \geq -2H\left|\widehat{\theta}_{t_h}(s_{t_h}, a_{t_h}, \delta_{t_h}) - \theta_{t_h}(s_{t_h}, a_{t_h}, \delta_{t_h})\right| - c_{A,2} H \sqrt{\frac{H\iota}{N_{t_h}(s_{t_h}, a_{t_h})}}.$$

It remains to bound the estimation error of $\widehat{\theta}_{t_h}(s_{t_h}, a_{t_h}, \delta_{t_h})$. Using the Hoeffding's inequality again, we obtain

$$\left| \widehat{\theta}_{t_h}(s_{t_h}, a_{t_h}, \delta_{t_h}) - \theta_{t_h}(s_{t_h}, a_{t_h}, \delta_{t_h}) \right| \leq c_\theta \sqrt{\frac{H\iota}{N_{t_h}(s_{t_h}, a_{t_h}, \delta_{t_h})}}.$$

Taking $c_A = \max\{c_{A,1}, c_{A,2}, c_\theta, 2\}$, we have

$$(A) \geq -c_A H \left( \sqrt{\frac{H\iota}{N_{t_h}(s_{t_h}, a_{t_h}, \delta_{t_h})}} + \sqrt{\frac{H\iota}{N_{t_h}(s_{t_h}, a_{t_h})}} \right).$$

With the choice of the bonus function, it can be checked that

$$\widetilde{Q}^k_{h,\mathrm{aug}}(\tau_h, a_h) - \widetilde{Q}^*_{h,\mathrm{aug}}(\tau_h, a_h) \geq (A) + b^k_h(\tau_h, a_h) \geq 0$$

with probability $1 - \gamma$ for any $(\tau_h, a_h)$. $\qquad\qquad\qquad\qquad\qquad\qquad\qquad\qquad\square$

**Regret analysis** In the sequel, we omit subscripts "aug" and tilde $\widetilde{\cdot}$ for simplicity. Thanks to Lemma B.1, we consider $(Q^k_h - Q^{\pi_k}_h)(\tau^k_h, a^k_h)$ as an upper bound of $(Q^*_h - Q^{\pi_k}_h)(\tau^k_h, a^k_h)$. We bound $(Q^k_h - Q^{\pi_k}_h)(\tau^k_h, a^k_h)$ by

$$\begin{aligned}
&(Q^k_h - Q^{\pi_k}_h)(\tau^k_h, a^k_h) \\
&\leq \left( [\widehat{\mathcal{P}}^k_h V^k_{h+1} - \mathcal{P}_h V^{\pi_k}_{h+1}] \right)(\tau^k_h, a^k_h) + b^k_h(\tau^k_h, a^k_h) \\
&\leq \left( [\widehat{\mathcal{P}}^k_h - \mathcal{P}_h] V^*_{h+1} \right)(\tau^k_h, a^k_h) + \left( [\widehat{\mathcal{P}}^k_h - \mathcal{P}_h][V^k_{h+1} - V^*_{h+1}] \right)(\tau^k_h, a^k_h) \\
&\quad + \left( \mathcal{P}_h[V^k_{h+1} - V^{\pi_k}_{h+1}] \right)(\tau^k_h, a^k_h) + b^k_h(\tau_h, a^k_h) \\
&\leq \underbrace{\left( [\widehat{\mathcal{P}}^k_h - \mathcal{P}_h][V^k_{h+1} - V^*_{h+1}] \right)(\tau^k_h, a^k_h)}_{(A)} + \left( \mathcal{P}_h[V^k_{h+1} - V^{\pi_k}_{h+1}] \right)(\tau^k_h, a^k_h) + 2b^k_h(\tau^k_h, a^k_h). \quad \text{(B.1)}
\end{aligned}$$

Similar to Lemma B.1, for $h \geq H$, we expand term $(A)$ into

$$\begin{aligned}
(A) &= \sum_{\tau_{h+1}} \left( \widehat{p}^k_h(\tau_{h+1}|\tau^k_h, a^k_h) - p_h(\tau_{h+1}|\tau^k_h, a^k_h) \right) [V^k_{h+1} - V^*_{h+1}](\tau_{h+1}) \\
&= \sum_{s_{t_h+1}} [V^k_{h+1} - V^*_{h+1}](\tau_{h+1}) \left( \widehat{p}^k_{t_h}(s_{t_h+1}|s^k_{t_h}, a^k_{t_h}) - p_{t_h}(s_{t_h+1}|s^k_{t_h}, a^k_{t_h}) \right). \quad \text{(B.2)}
\end{aligned}$$

On the other hand, for $h \leq H$, the decomposition of term $(A)$ is more complicated. We have

$$\begin{aligned}
(A) &= \sum_{\tau_{h+1}} \left( \widehat{p}^k_h(\tau_{h+1}|\tau^k_h, a^k_h) - p_h(\tau_{h+1}|\tau^k_h, a^k_h) \right) [V^k_{h+1} - V^*_{h+1}](\tau_{h+1}) \\
&= \underbrace{\sum_{\tau_{h+1}} [V^k_{h+1} - V^*_{h+1}](\tau_{h+1}) \left( \widehat{p}^k_h(\tau_{h+1}|\tau^k_h, a^k_h) - p_h(\tau_{h+1}|\tau^k_h, a^k_h) \right) \mathbb{1}\{\delta_{t_h+1} = 0\} \mathbb{1}\{t_{h+1} = t^k_h + 1\}}_{(A_1)} \\
&\quad + \underbrace{\sum_{\tau_{h+1}} [V^k_{h+1} - V^*_{h+1}](\tau_{h+1}) \left( \widehat{p}^k_h(\tau_{h+1}|\tau^k_h, a^k_h) - p_h(\tau_{h+1}|\tau^k_h, a^k_h) \right) \mathbb{1}\{\delta_{t_h+1} = \delta_{t^k_h} + 1\} \mathbb{1}\{t_{h+1} = t^k_h\}}_{(A_2)}.
\end{aligned}$$

Term $(A_2)$ can be directly bounded by

$$\begin{aligned}
(A_2) &\leq H \left| \widehat{\theta}^k_{t_h}(s^k_{t_h}, a^k_{t_h}, \delta^k_{t_h}) - \theta_{t_h}(s^k_{t_h}, a^k_{t_h}, \delta^k_{t_h}) \right| \\
&\leq c_\theta H \sqrt{\frac{H\iota}{N^k_{t_h}(s^k_{t_h}, a^k_{t_h}, \delta^k_{t_h})}}
\end{aligned}$$

with probability $1 - \gamma$. To bound $(A_1)$, we have

$$(A_1) = \sum_{s_{t_{h+1}}} [V_{h+1}^k - V_{h+1}^*](\tau_{h+1}) \Bigg( \Big(1 - \widehat{\theta}_{t_h}^k(s_{t_h}^k, a_{t_h}^k, \delta_{t_h}^k)\Big) \widehat{p}_{t_h}^k(s_{t_{h+1}} | s_{t_h}^k, a_{t_h}^k)$$

$$- \Big(1 - \theta_{t_h}(s_{t_h}^k, a_{t_h}^k, \delta_{t_h}^k)\Big) p_{t_h}(s_{t_{h+1}} | s_{t_h}^k, a_{t_h}^k) \Bigg)$$

$$= \sum_{s_{t_{h+1}}} [V_{h+1}^k - V_{h+1}^*](\tau_{h+1}) \Big( \Big(1 - \widehat{\theta}_{t_h}^k(s_{t_h}^k, a_{t_h}^k, \delta_{t_h}^k)\Big) - \Big(1 - \theta_{t_h}(s_{t_h}^k, a_{t_h}^k, \delta_{t_h}^k)\Big) \Big) \widehat{p}_{t_h}^k(s_{t_{h+1}} | s_{t_h}^k, a_{t_h}^k)$$

$$+ \sum_{s_{t_{h+1}}} [V_{h+1}^k - V_{h+1}^*](\tau_{h+1}) \Big(1 - \theta_{t_h}(s_{t_h}^k, a_{t_h}^k, \delta_{t_h}^k)\Big) \big( \widehat{p}_{t_h}^k(s_{t_{h+1}} | s_{t_h}^k, a_{t_h}^k) - p_{t_h}(s_{t_{h+1}} | s_{t_h}^k, a_{t_h}^k) \big)$$

$$\leq \Big(1 - \theta_{t_h}(s_{t_h}^k, a_{t_h}^k, \delta_{t_h}^k)\Big) \sum_{s_{t_{h+1}}} [V_{h+1}^k - V_{h+1}^*](\tau_{h+1}) \big( \widehat{p}_{t_h}^k(s_{t_{h+1}} | s_{t_h}^k, a_{t_h}^k) - p_{t_h}(s_{t_{h+1}} | s_{t_h}^k, a_{t_h}^k) \big)$$

$$+ H \Big| \widehat{\theta}_{t_h}^k(s_{t_h}^k, a_{t_h}^k, \delta_{t_h}^k) - \theta_{t_h}(s_{t_h}^k, a_{t_h}^k, \delta_{t_h}^k) \Big|$$

$$\leq \Big(1 - \theta_{t_h}(s_{t_h}^k, a_{t_h}^k, \delta_{t_h}^k)\Big) \sum_{s_{t_{h+1}}} [V_{h+1}^k - V_{h+1}^*](\tau_{h+1}) \big( \widehat{p}_{t_h}^k(s_{t_{h+1}} | s_{t_h}^k, a_{t_h}^k) - p_{t_h}(s_{t_{h+1}} | s_{t_h}^k, a_{t_h}^k) \big)$$

$$+ c_\theta H \sqrt{\frac{H \iota}{N_{t_h}^k(s_{t_h}^k, a_{t_h}^k, \delta_{t_h}^k)}}.$$

Putting $(A_1)$ and $(A_2)$ together, we obtain

$$(A) \leq \Big(1 - \theta_{t_h}(s_{t_h}^k, a_{t_h}^k, \delta_{t_h}^k)\Big) \sum_{s_{t_{h+1}}} [V_{h+1}^k - V_{h+1}^*](\tau_{h+1}) \big( \widehat{p}_{t_h}^k(s_{t_{h+1}} | s_{t_h}^k, a_{t_h}^k) - p_{t_h}(s_{t_{h+1}} | s_{t_h}^k, a_{t_h}^k) \big)$$

$$+ 2 c_\theta H \sqrt{\frac{H \iota}{N_{t_h}^k(s_{t_h}^k, a_{t_h}^k, \delta_{t_h}^k)}}. \tag{B.3}$$

In both (B.2) and (B.3) for different ranges of $h$, we apply the Bernstein inequality (Lemma D.1) to derive

$$\sum_{s_{t_{h+1}}} [V_{h+1}^k - V_{h+1}^*](\tau_{h+1}) \big( \widehat{p}_{t_h}^k(s_{t_{h+1}} | s_{t_h}^k, a_{t_h}^k) - p_{t_h}(s_{t_{h+1}} | s_{t_h}^k, a_{t_h}^k) \big)$$

$$\leq c \cdot \sum_{s_{t_{h+1}}} [V_{h+1}^k - V_{h+1}^*](\tau_{h+1}) \left[ \sqrt{\frac{p_{t_h}(s_{t_{h+1}} | s_{t_h}^k, a_{t_h}^k) \iota}{N_{t_h}^k(s_{t_h}^k, a_{t_h}^k)}} + \frac{\iota}{N_{t_h}^k(s_{t_h}^k, a_{t_h}^k)} \right]$$

$$\overset{(i)}{\leq} c \cdot \sum_{s_{t_{h+1}}} [V_{h+1}^k - V_{h+1}^*](\tau_{h+1}) \left[ \frac{p_{t_h}(s_{t_{h+1}} | s_{t_h}^k, a_{t_h}^k)}{2cH} + \frac{(2cH + 1)\iota}{N_{t_h}^k(s_{t_h}^k, a_{t_h}^k)} \right]$$

$$\leq c \cdot \left( \frac{SH(2cH + 1)\iota}{N_{t_h}^k(s_{t_h}^k, a_{t_h}^k)} + \frac{1}{2cH} \sum_{s_{t_{h+1}}} [V_{h+1}^k - V_{h+1}^*](\tau_{h+1}) p_{t_h}(s_{t_{h+1}} | s_{t_h}^k, a_{t_h}^k) \right), \tag{B.4}$$

where inequality $(i)$ follows from $\sqrt{ab} \leq a + b$. Substituting (B.4) into (B.2), for $h \geq H$, we deduce

$$(A) \leq \frac{1}{2H} \sum_{s_{t_{h+1}}} [V_{h+1}^k - V_{h+1}^*](\tau_{h+1}) p_{t_h}(s_{t_{h+1}} | s_{t_h}^k, a_{t_h}^k) + \frac{cSH(2cH + 1)\iota}{N_{t_h}^k(s_{t_h}^k, a_{t_h}^k)}$$

$$\overset{(i)}{\leq} \frac{1}{2H} \big( \mathcal{P}_h[V_{h+1}^k - V_{h+1}^{\pi_k}] \big) (\tau_h^k, a_h^k) + c' \frac{mSH^2\iota}{N_{t_h}^k(s_{t_h}^k, a_{t_h}^k)},$$

where $c'$ is a sufficiently large constant. By the same reasoning, substituting (B.4) into (B.3), for $h < H$, we have

$$(A) \leq \frac{1}{2H} \Big(1 - \theta_{t_h}(s_{t_h}^k, a_{t_h}^k, \delta_{t_h}^k)\Big) \sum_{s_{t_{h+1}}} [V_{h+1}^k - V_{h+1}^*](\tau_{h+1}) p_{t_h}(s_{t_{h+1}} | s_{t_h}^k, a_{t_h}^k) + \frac{cSH(2cH + 1)\iota}{N_{t_h}^k(s_{t_h}^k, a_{t_h}^k)}$$

$$+ 2c_\theta H \sqrt{\frac{H\iota}{N_{t_h}^k(s_{t_h}^k, a_{t_h}^k, \delta_{t_h}^k)}}$$

$$\overset{(i)}{\leq} \frac{1}{2H} \left( \mathcal{P}_h[V_{h+1}^k - V_{h+1}^{\pi_k}] \right)(\tau_h^k, a_h^k) + c' \frac{SH^2\iota}{N_{t_h}^k(s_{t_h}^k, a_{t_h}^k)} + 2c_\theta H \sqrt{\frac{H\iota}{N_{t_h}^k(s_{t_h}^k, a_{t_h}^k, \delta_{t_h}^k)}}.$$

We denote $\zeta_h^k = c' \frac{SH^2\iota}{N_{t_h}^k(s_{t_h}^k, a_{t_h}^k)}$. Now we have a unified upper bound on $(A)$ for any $h \in [1, \widetilde{H}]$ as

$$(A) \leq \frac{1}{2H} \left( \mathcal{P}_h[V_{h+1}^k - V_{h+1}^{\pi_k}] \right)(\tau_h^k, a_h^k) + \zeta_h^k + 2c_\theta H \sqrt{\frac{H\iota}{N_{t_h}^k(s_{t_h}^k, a_{t_h}^k, \delta_{t_h}^k)}}. \tag{B.5}$$

Substituting (B.5) back into (B.1), we have

$$\left( V_h^k - V_h^{\pi_k} \right)(\tau_h^k) = \left( Q_h^k - Q_h^{\pi_k} \right)(\tau_h^k, a_h^k)$$

$$\leq \left( 1 + \frac{1}{2H} \right) \left( \mathcal{P}_h \left[ V_h^k - V_h^{\pi_k} \right] \right)(\tau_h^k, a_h^k) + \zeta_h^k + 2b_h^k + 2c_\theta H \sqrt{\frac{H\iota}{N_{t_h}^k(s_{t_h}^k, a_{t_h}^k, \delta_{t_h}^k)}}.$$

We further denote $\xi_h^k = \left( \mathcal{P}_h \left[ V_h^k - V_h^{\pi_k} \right] \right)(\tau_h^k, a_h^k) - \left[ V_{h+1}^k - V_{h+1}^{\pi_k} \right](\tau_{h+1}^k)$ and rewrite $\left( V_h^k - V_h^{\pi_k} \right)(\tau_h^k)$ as

$$\left( V_h^k - V_h^{\pi_k} \right)(\tau_h^k) \leq \left( 1 + \frac{1}{2H} \right) \left( \left[ V_{h+1}^k - V_{h+1}^{\pi_k} \right](\tau_{h+1}^k) + \xi_h^k \right) + \zeta_h^k + 2b_h^k + 2c_\theta H \sqrt{\frac{H\iota}{N_{t_h}^k(s_{t_h}^k, a_{t_h}^k, \delta_{t_h}^k)}}.$$

Recall $\widetilde{H} = 2H$. Using a recursive summation argument, we deduce

$$\left( V_1^k - V_1^{\pi_k} \right)(\tau_1^k) \leq \sum_{h=1}^{\widetilde{H}} \left( 1 + \frac{1}{2H} \right)^h \left( \xi_h^k + \zeta_h^k + 2b_h^k + 2c_\theta H \sqrt{\frac{H\iota}{N_{t_h}^k(s_{t_h}^k, a_{t_h}^k, \delta_{t_h}^k)}} \right)$$

$$\leq e \sum_{h=1}^{2H} \left( \xi_h^k + \zeta_h^k + 2b_h^k + 2c_\theta H \sqrt{\frac{H\iota}{N_{t_h}^k(s_{t_h}^k, a_{t_h}^k, \delta_{t_h}^k)}} \right).$$

As a consequence, the total regret is bounded by

$$\texttt{Regret}(K) \leq e \sum_{k=1}^{K} \sum_{h=1}^{2H} \left( \xi_h^k + \zeta_h^k + 2b_h^k + 2c_\theta H \sqrt{\frac{H\iota}{N_{t_h}^k(s_{t_h}^k, a_{t_h}^k, \delta_{t_h}^k)}} \right). \tag{B.6}$$

We need to sum over $\zeta_h^k, \xi_h^k, b_h^k$. Consider $\zeta_h^k$ first. We have

$$\sum_{k=1}^{K} \sum_{h=1}^{2H} \zeta_h^k = c' \sum_{k=1}^{K} \sum_{h=1}^{2H} \frac{SH^2\iota}{N_{t_h}^k(s_{t_h}^k, a_{t_h}^k)}$$

$$\overset{(i)}{\leq} c'H \sum_{k=1}^{K} \sum_{h=1}^{H} \frac{SH^2\iota}{N_h^k(s_h^k, a_h^k)}$$

$$\overset{(ii)}{\leq} c_\zeta H^4 S^2 A \iota^2, \tag{B.7}$$

where inequality $(i)$ invokes the fact that $t_h$ only takes value in $\{1, \ldots, H\}$ and each $N_{t_h}^k(s_{t_h}^k, a_{t_h}^k)$ is repeated at most $H$ times, and inequality $(ii)$ follows from the pigeon-hole argument in Azar et al. [2017].

Next we bound the summation over $\xi_h^k$. This is a martingale difference sequence. We apply Azuma-Hoeffding's inequality (Lemma D.3) with $n = 2H$ and $c_i = 4H$ to obtain

$$\sum_{k=1}^{K} \sum_{h=1}^{2H} \xi_h^k \leq c_\xi \sqrt{KH^4\iota}. \tag{B.8}$$

The additional $H$ dependence above comes from a union bound over $\widetilde{\mathcal{S}}_{\mathrm{aug}} \times \mathcal{A}$. Lastly, we tackle the summation over bonus functions $b_h^k$. We have

$$
\begin{aligned}
\sum_{k=1}^{K} \sum_{h=1}^{2H} b_h^k &= \sum_{k=1}^{K} \sum_{h=1}^{2H} c_A H \sqrt{\frac{H\iota}{N_{t_h}^k(s_{t_h}, a_{t_h})}} \\
&\leq c_A H \sum_{k=1}^{K} \sum_{h=1}^{H} H \sqrt{\frac{H\iota}{N_{t_h}^k(s_{t_h}, a_{t_h})}} \\
&\leq c_b H^{7/2} \sqrt{SAK\iota}.
\end{aligned}
\tag{B.9}
$$

Putting (B.7), (B.8) and (B.9) together, we deduce

$$
\texttt{Regret}(K) \leq c \left( H^{7/2}\sqrt{SAK\iota} + H^4 S^2 A\iota^2 + \sqrt{H^4 K\iota} \right) + 2ec_\theta H \sum_{k=1}^{K} \sum_{h=1}^{2H} \sqrt{\frac{H\iota}{N_{t_h}^k(s_{t_h}^k, a_{t_h}^k, \delta_{t_h}^k)}}
$$

for some constant $c$. To this end, the only remaining task is to find $\sum_{k=1}^{K} \sum_{h=1}^{2H} \sqrt{\frac{1}{N_{t_h}^k(s_{t_h}^k, a_{t_h}^k, \delta_{t_h}^k)}}$, which undergoes a similar argument as the bonus summation. We have

$$
\begin{aligned}
\sum_{k=1}^{K} \sum_{h=1}^{2H} \sqrt{\frac{1}{N_{t_h}^k(s_{t_h}^k, a_{t_h}^k, \delta_{t_h}^k)}} &\leq H \sum_{k=1}^{K} \sum_{h=1}^{H} \sqrt{\frac{1}{N_h^k(s_h^k, a_h^k, \delta_h^k)}} \\
&= H \sum_{(h,s,a,\delta)} \sum_{i=1}^{N_h^K(s,a,\delta)} \sqrt{\frac{1}{i}} \\
&\overset{(i)}{\leq} 2H \sum_{\delta} \sum_{(h,s,a)} \sqrt{N_h^K(s, a, \delta)} \\
&\overset{(ii)}{\leq} 2H \sum_{\delta} \sqrt{SAKH} \\
&\overset{(iii)}{\leq} 2H^2 \sqrt{SAKH},
\end{aligned}
\tag{B.10}
$$

where inequality $(i)$ invokes $\sum_{i=1}^{n} 1/\sqrt{i} \leq 2\sqrt{n}$, inequality $(ii)$ follows from Cauchy-Schwarz, and inequality $(iii)$ uses the fact that $\delta$ is bounded by $H$. Plugging (B.10) into the regret bound, we obtain the desired result

$$
\texttt{Regret}(K) \leq c \left( H^4 \sqrt{SAK\iota} + H^4 S^2 A\iota^2 + \sqrt{H^4 K\iota} \right)
$$

with probability $1 - \gamma$. Absorbing $\sqrt{H^4 K\iota}$ into $H^4 \sqrt{SAK\iota}$ yields the bound in Theorem 4.1. $\square$

## B.2 Regret under bounded delay

To obtain the regret bound under bounded delay, we only need to modify several steps in the proof of Theorem 4.1. Specifically, in Lemma B.1, we replace $H$ by $D + 1$ in the square root, so that the bonus function becomes

$$
b_h^k(\tau_h, a_h) = c_A H \left( \sqrt{\frac{(D+1)\iota}{N_{t_h}(s_{t_h}, a_{t_h}, \delta_{t_h})}} + \sqrt{\frac{(D+1)\iota}{N_{t_h}(s_{t_h}, a_{t_h})}} \right).
$$

The optimism still holds, since the expanded state space contains at most $D$ historical actions.

The second modification is to observe that since the delay is bounded by $D$, each counting number $N_h^k$ only repeats at most $D + 1$ times. In this way, (B.7) becomes

$$
\sum_{k=1}^{K} \sum_{h=1}^{2H} \zeta_h^k \leq c_\zeta (D+1) H^3 S^2 A\iota^2.
$$

(B.8) and (B.9) are replaced by

$$\sum_{k=1}^{K}\sum_{h=1}^{2H}\xi_h^k \le c_\xi(D+1)\sqrt{K(D+1)H^3\iota} \quad \text{and} \quad \sum_{k=1}^{K}\sum_{h=1}^{2H}b_h^k \le c_b(D+1)^{3/2}\sqrt{H^3SAK\iota},$$

respectively. Lastly, we also have

$$\sum_{k=1}^{K}\sum_{h=1}^{2H}\sqrt{\frac{1}{N_{t_h}^k\left(s_{t_h}^k,a_{t_h}^k,\delta_{t_h}^k\right)}} \le 2(D+1)^2\sqrt{SAKH}.$$

Putting these updated upper bounds together and substituting into $\texttt{Regret}(K)$, we deduce

$$\texttt{Regret}(K) \le c\left((D+1)^{5/2}\sqrt{H^3SAK\iota} + H^4S^2A\iota^2\right).$$

## B.3 Statement and proof of Proposition B.2

**Proposition B.2.** In the setup of Section 4.2, we have

$$\texttt{gap}(s_1) \le \sum_{h=1}^{H}\Bigg[\underbrace{\int_\tau \left(\mathbb{E}_{s\sim\mathfrak{b}_h(\cdot|\tau)}[\max_a r_h(s,a)] - \max_a \mathbb{E}_{s\sim\mathfrak{b}_h(\cdot|\tau)}[r_h(s,a)]\right)\left(\rho_h^{\pi_{\text{delay}}^*}\wedge\rho_h^{\pi_{\text{nodelay}}^*}\right)(\tau)\mathrm{d}\tau}_{\mathcal{E}_1}$$

$$+ 2\underbrace{\|\rho_h^{\pi_{\text{nodelay}}^*} - \rho_h^{\pi_{\text{delay}}^*}\|_{\text{TV}}}_{\mathcal{E}_2}\Bigg].$$

where $\rho_h^{\pi_{\text{nodelay}}^*}$ and $\rho_h^{\pi_{\text{delay}}^*}$ are visitation measures induced by $\pi_{\text{nodelay}}^*$ and $\pi_{\text{delay}}^*$, respectively.

Term $\mathcal{E}_1$ is strictly larger than zero due to the convexity of the max operation. Term $\mathcal{E}_2$ accounts for the difference in the visitation measure. When the original MDP has deterministic transitions, we can check that $\mathcal{E}_1$ is zero, since the expectation over $s$ is concentrated on a singleton that can be inferred from history. Hence, the visitation measures are also identical, which implies $V_{1,\text{nodelay}}^*(s_1) - V_{1,\text{delay}}^*(s_1) = 0$. On the contrary, when $\mathfrak{b}_h(\cdot|\tau)$ is evenly spread, meaning that the entropy of $\mathfrak{b}_h$ is high, we potentially suffer from a large performance drop, in that, inferring the current state is difficult.

*Proof of Proposition B.2.* Let $\tau_1,\ldots,\tau_H$ denote the states observed in the delayed environment. Since $\pi_{\text{nodelay}}^*$ is greedy and Markov, we obtain

$$V_{1,\text{nodelay}}^*(s_1) = \mathbb{E}^{\pi_{\text{nodelay}}^*}\left[\sum_{h=1}^{H-1}r_h(s_h,a_h)|s_1\right] + \mathbb{E}^{\pi_{\text{nodelay}}^*}\left[\mathbb{E}[r_H(s_H,a_H)|\tau_H]|s_1\right]$$

$$= \mathbb{E}^{\pi_{\text{nodelay}}^*}\left[\sum_{h=1}^{H-1}r_h(s_h,a_h)|s_1\right] + \mathbb{E}^{\pi_{\text{nodelay}}^*}\left[\sum_s \mathfrak{b}_H(s|\tau_H)\max_a r_H(s,a)|s_1\right].$$

Recursively applying the above argument, we deduce

$$V_{1,\text{nodelay}}^*(s_1) = \mathbb{E}^{\pi_{\text{nodelay}}^*}\left[\sum_{h=1}^{H}\sum_s \mathfrak{b}_h(s|\tau_h)\max_a r_h(s,a)|s_1\right].$$

We also rewrite $V_{1,\text{delay}}^*(s_1)$ as

$$V_{1,\text{delay}}^*(s_1) = \mathbb{E}^{\pi_{\text{delay}}^*}\left[\sum_{h=1}^{H-1}r_h(s_h,a_h)|s_1\right] + \mathbb{E}^{\pi_{\text{delay}}^*}\left[\mathbb{E}[r_H(s_H,a_H)|\tau_H]|s_1\right]$$

$$= \mathbb{E}^{\pi_{\text{delay}}^*}\left[\sum_{h=1}^{H-1}r_h(s_h,a_h)|s_1\right] + \mathbb{E}^{\pi_{\text{delay}}^*}\left[\max_a \sum_s \mathfrak{b}_H(s|\tau_H)r_H(s,a)|s_1\right]$$

$$= ...$$

$$= \mathbb{E}^{\pi^*_{\text{delay}}}\left[\sum_{h=1}^{H}\max_a \sum_s \mathfrak{b}_h(s|\tau_h)r_h(s,a)|s_1\right].$$

Then we write the difference between $V^*_{1,\text{nodelay}}(s_1)$ and $V^*_{1,\text{delay}}(s_1)$ as

$$V^*_{1,\text{nodelay}}(s_1) - V^*_{1,\text{delay}}(s_1)$$

$$= \sum_{h=1}^{H}\left(\int_\tau \sum_s \max_a \mathfrak{b}_h(s|\tau)r_h(s,a)\rho_h^{\pi^*_{\text{nodelay}}}(\tau)\mathrm{d}\tau - \int_\tau \max_a \sum_s \mathfrak{b}_h(s|\tau)r_h(s,a)\rho_h^{\pi^*_{\text{delay}}}(\tau)\mathrm{d}\tau\right)$$

$$= \sum_{h=1}^{H}\left(\int_\tau \sum_s \max_a \mathfrak{b}_h(s|\tau)r_h(s,a)\rho_h^{\pi^*_{\text{nodelay}}}(\tau)\mathrm{d}\tau - \int_\tau \max_a \sum_s \mathfrak{b}_h(s|\tau)r_h(s,a)\rho_h^{\pi^*_{\text{nodelay}}}(\tau)\mathrm{d}\tau\right.$$

$$\left. + \int_\tau \max_a \sum_s \mathfrak{b}_h(s|\tau)r_h(s,a)\rho_h^{\pi^*_{\text{nodelay}}}(\tau)\mathrm{d}\tau - \int_\tau \max_a \sum_s \mathfrak{b}_h(s|\tau)r_h(s,a)\rho_h^{\pi^*_{\text{delay}}}(\tau)\mathrm{d}\tau\right)$$

$$\leq \sum_{h=1}^{H}\left[\int_\tau\left(\mathbb{E}_{s\sim\mathfrak{b}_h(\cdot|\tau)}[\max_a r_h(s,a)] - \max_a \mathbb{E}_{s\sim\mathfrak{b}_h(\cdot|\tau)}[r_h(s,a)]\right)\rho_h^{\pi^*_{\text{nodelay}}}(\tau)\mathrm{d}\tau + 2\|\rho_h^{\pi^*_{\text{nodelay}}} - \rho_h^{\pi^*_{\text{delay}}}\|_{\text{TV}}\right].$$

We also have

$$V^*_{1,\text{nodelay}}(s_1) - V^*_{1,\text{delay}}(s_1)$$

$$= \sum_{h=1}^{H}\left(\int_\tau \sum_s \max_a \mathfrak{b}_h(s|\tau)r_h(s,a)\rho_h^{\pi^*_{\text{nodelay}}}(\tau)\mathrm{d}\tau - \int_\tau \sum_s \max_a \mathfrak{b}_h(s|\tau)r_h(s,a)\rho_h^{\pi^*_{\text{delay}}}(\tau)\mathrm{d}\tau\right.$$

$$\left. + \int_\tau \sum_s \max_a \mathfrak{b}_h(s|\tau)r_h(s,a)\rho_h^{\pi^*_{\text{delay}}}(\tau)\mathrm{d}\tau - \int_\tau \max_a \sum_s \mathfrak{b}_h(s|\tau)r_h(s,a)\rho_h^{\pi^*_{\text{delay}}}(\tau)\mathrm{d}\tau\right)$$

$$\leq \sum_{h=1}^{H}\left[\int_\tau\left(\mathbb{E}_{s\sim\mathfrak{b}_h(\cdot|\tau)}[\max_a r_h(s,a)] - \max_a \mathbb{E}_{s\sim\mathfrak{b}_h(\cdot|\tau)}[r_h(s,a)]\right)\rho_h^{\pi^*_{\text{delay}}}(\tau)\mathrm{d}\tau + 2\|\rho_h^{\pi^*_{\text{nodelay}}} - \rho_h^{\pi^*_{\text{delay}}}\|_{\text{TV}}\right].$$

Combining the above two inequalities, we obtain

$$V^*_{1,\text{nodelay}}(s_1) - V^*_{1,\text{delay}}(s_1)$$

$$\leq \sum_{h=1}^{H}\left[\int_\tau\left(\mathbb{E}_{s\sim\mathfrak{b}_h(\cdot|\tau)}[\max_a r_h(s,a)] - \max_a \mathbb{E}_{s\sim\mathfrak{b}_h(\cdot|\tau)}[r_h(s,a)]\right)\left(\rho_h^{\pi^*_{\text{delay}}} \wedge \rho_h^{\pi^*_{\text{nodelay}}}\right)(\tau)\mathrm{d}\tau\right.$$

$$\left. + 2\|\rho_h^{\pi^*_{\text{nodelay}}} - \rho_h^{\pi^*_{\text{delay}}}\|_{\text{TV}}\right].$$

The proof is complete. $\qquad\qquad\qquad\qquad\qquad\qquad\qquad\qquad\qquad\qquad\qquad\qquad\square$

### B.4  Proof of Proposition 4.2

*Proof.* We construct an MDP instance $(\mathcal{S}, \mathcal{A}, H, R, P)$ for $H > d$ as follows. Let $\mathcal{S} = \{1, 2\}$ and $\mathcal{A} = \{a_1, a_2\}$. For the reward function, we have

$$r_h(s,a) = \begin{cases} 1 & \text{if } a = a_s \text{ and } h = d+1 \\ 0 & \text{otherwise} \end{cases}.$$

The reward is nonzero only at time $d+1$. The transition probabilities are defined as

$$p_h(s'|s,a) = \begin{cases} \frac{1}{2} & \text{if } h = d+1 \\ 1 & \text{if } h \neq d+1 \text{ and } s' = s \\ 0 & \text{otherwise} \end{cases}.$$

The transition probability at step $d+1$ says that $s'$ is uniform regardless of the previous state and action. Suppose a uniform initial distribution on $s_1$. We first show that if the constant delay equals $d$, then there exists a policy $\pi^{*,d}$ achieving maximal value. Indeed, the policy is chosen as

$$\pi^{*,d}_h(\cdot|\{s_{h-d}, \mathbf{a}_{h-d:h-1}\}) = \begin{cases} a_{s_{h-d}} & \text{if } h = d+1 \\ \text{Uniform}(\mathcal{A}) & \text{if } h \neq d+1. \end{cases}$$

It is straightforward to check that $\pi^{*,d}$ is optimal, since at step $d+1$, $s_1$ is revealed and the policy takes the optimal action $a_{s_1}$ to obtain reward 1.

On the other hand, if the constant delay equals $d+1$, then any policy suffers from a constant performance degradation. To see this, in a single trajectory, since the starting state is only revealed at time $d+2$, the policy at time $d+1$ cannot exploit the information of the initial state. Therefore, any policy coincides with the best action with probability $\frac{1}{2}$. For $K$ episodes, with probability $1-\gamma$, the total reward of any policy $\pi \in \Pi_{\text{exec}}$ is bounded by

$$\sum_{k=1}^{K} V_1^{\pi}(s_1^k) \leq \frac{1}{2}K + \sqrt{\frac{K}{2}\log\frac{1}{\gamma}},$$

due to Hoeffding's inequality. As a result, the performance drop is at least

$$\text{gap}(K) \geq \frac{1}{2} - \sqrt{\frac{1}{2K}\log\frac{1}{\gamma}}.$$

$\square$

## C  Omitted proofs in Section 5

### C.1  Proof of Proposition 5.1

*Proof.* We first show that the ground-truth transition probabilities $p_h^{\theta^*}$ belongs to $\mathcal{B}_k$ with high probability. By Theorem 2.2 in Weissman et al. [2003] (see also Equation (44) in Jaksch et al. [2010]), at the $k$-th episode, for any fixed $(s, a, h)$, we have

$$\mathbb{P}\left(\|\widehat{p}_h^k(\cdot|s,a) - p_h^{\theta^*}(\cdot|s,a)\|_{\text{TV}} \geq t\right) \leq (2^S - 2)\exp\left(-\frac{N_h^k(s,a)t^2}{2}\right).$$

Setting $t = c\sqrt{\frac{S\iota}{N_h^k(s,a)}}$ for some constant $c$ ensures that

$$\|\widehat{p}_h^k(\cdot|s,a) - p_h^{\theta^*}(\cdot|s,a)\|_{\text{TV}} \leq c\sqrt{\frac{S\iota}{N_h^k(s,a)}}$$

holds over any $(s,a,h,k)$ with probability $1-\gamma$. As a consequence, the event $p_h^{\theta^*}(\cdot|s,a) \in \mathcal{B}^k$ holds with probability $1-\gamma$ over all $(s,a,h,k)$.

Conditioned on the high probability event $p_h^{\theta^*} \in \mathcal{B}^k$ for all $(h,s,a)$, we have by standard performance difference arguments that

$$\sum_{k=1}^{K}\max_{\pi\in\Pi_{\text{exec}}} V_{\theta^*}^{\pi}(s_1^k) - V_{\theta^*}^{\pi^k}(s_1^k) \overset{(i)}{\leq} \sum_{k=1}^{K} V_{\theta^k}^{\pi^k}(s_1^k) - V_{\theta^*}^{\pi^k}(s_1^k)$$

$$\overset{(ii)}{=} \sum_{k=1}^{K}\sum_{h=1}^{H}\mathbb{E}_{\theta^*}^{\pi^k}\left[\left\langle(\mathbb{P}_h^{\theta^k} - \mathbb{P}_h^{\theta^*})(\cdot|s_h, a_h), V_{\theta^k,h+1}^{\pi^k}(\cdot)\right\rangle\right]$$

$$\leq \sum_{h=1}^{H}\sum_{k=1}^{K}\mathbb{E}_{\theta^*}^{\pi^k}\left[c\sqrt{\frac{H^2 S\iota}{N_h^k(s_h, a_h)}} \wedge H\right]$$

$$\overset{(iii)}{\leq} \sum_{h=1}^{H}\sum_{k=1}^{K}c'\sqrt{\frac{H^2 S\iota}{N_h^k(s_h^k, a_h^k)}} + H\sqrt{H^2 K\iota}$$

$$\overset{(iv)}{\leq} c'\left(\left\lceil \frac{\log \frac{HK}{\gamma}}{-\log(1-\lambda_0^2)}\right\rceil \sqrt{H^2 S\iota \cdot SAHK} + \sqrt{H^4 K\iota}\right)$$

$$\leq c'\left(\left\lceil \frac{1}{-\log(1-\lambda_0^2)}\right\rceil \sqrt{H^3 S^2 AK\iota^3} + \sqrt{H^4 K\iota}\right),$$

where inequality $(i)$ follows from the valid optimism since Line 4 in Algorithm 3 is taken over double maximization, equality $(ii)$ recursively expands the value function and $\langle \cdot, \cdot \rangle$ denotes the inner product, inequality $(iii)$ invokes Azuma-Hoeffding's inequality, and inequality $(iv)$ invokes Lemma C.2. $\quad\square$

## C.2 Algorithm and proof of Theorem 5.2

---

**Algorithm 4** Policy learning for MDPs with missing observations

---

1: **Input**: Horizon $H$.
2: **Init**: $V_{H+1}(\tau) = 0$ and $Q_H(\tau, a) = H$ for any $\tau, a$, data set $\mathcal{D}^0 = \emptyset$, initial policy $\pi^0$.
3: **for** episode $k = 1, \ldots, K$ **do**
4: $\quad$ Execute policy $\pi^{k-1}$.
5: $\quad$ After the episode ends, collect data $\mathcal{D}^k = \mathcal{D}^{k-1} \cup \{(s_h^k, a_h^k, r_h^k)\}_{h=1}^H$.
6: $\quad$ On data set $\mathcal{D}^k$, compute counting numbers

$$N_h^k(\tau_h, a_h) = \sum_{j=1}^k \mathbb{1}\{\tau_h^k = \tau_h, a_h^k = a_h, s_{h+1}^k \neq \emptyset\} \quad \text{and} \quad N_{h,\lambda}^k = \sum_{j=1}^k \mathbb{1}\{s_h^k = \emptyset\}.$$

7: $\quad$ Estimate transition probabilities and delay distributions via

$$\widehat{p}_h^k(s_{h+1}|\tau_h, a_h) = \frac{N_h^k(\tau_h, a_h, s_{h+1})}{N_h^k(\tau_h, a_h)} \quad \text{and} \quad \widehat{\lambda}_h^k = N_{h,\lambda}^k/k.$$

8: $\quad$ Set bonus function as

$$b_h^k(\tau_h, a_h) = cH\left(\sqrt{\frac{H\iota}{N_h^k(\tau_h, a_h)}} + \sqrt{\frac{\iota}{k}}\right)$$

$\quad\quad$ for $\iota = \log \frac{SAKH}{\gamma}$ and $c$ sufficiently large.
9: $\quad$ Run optimistic value iteration in $\texttt{MDP}_{\text{aug}}$ for $H$ steps and obtain $\pi^k \in \Pi_{\text{exec}}$.
10: **end for**
11: **Return**: Learned policy $\pi^k$ for $k = 1, \ldots, K$.

---

We remark that similar to delayed MDPs, in Line 9 the planning is on $\texttt{MDP}_{\text{aug}}$ and the obtained policy is executable given any $\tau \in \mathcal{S}_{\text{aug}}$ when state observation is missed. Therefore, the planning complexity is $SA^H$. Different from Algorithm 2, the bonus function here depends on multi-step transitions, in that missing observations are permanently lost.

*Proof of Theorem 5.2.* The proof utilizes similar steps as Theorem 4.1, with an extra care on the summation of bonus functions.

**Valid optimism** We verify the choice of bonus functions leads to a valid optimism.

**Lemma C.1.** Given any failure probability $\gamma < 1$, we set bonus functions as

$$b_h^k(\tau_h, a_h) = cH\left(\sqrt{\frac{H\iota}{N_h^k(\tau_h, a_h)}} + \sqrt{\frac{\iota}{k}}\right) \quad \text{with} \quad \iota = \log\left(\frac{SAHK}{\gamma}\right).$$

Then with probability $1 - \gamma$, it holds

$$Q_{h,\text{aug}}^k(\tau_h, a_h) \geq Q_{h,\text{aug}}^*(\tau_h, a_h), \quad V_{h,\text{aug}}^k(\tau_h) \geq V_{h,\text{aug}}^*(\tau_h) \quad \text{for any} \quad (k, h, \tau_h, a_h).$$

*Proof of Lemma C.1.* In the proof, we omit subscript "aug" for simplicity. We use backward induction on time $h$ again. The base case of $H + 1$ holds immediately due to the initial value of $V_{H+1,\text{aug}}$. Suppose at time $h + 1$, the assertion holds. Then for time $h$, if $Q_{h,\text{aug}} = H$, the assertion holds trivially. Otherwise, we have

$$Q_h(\tau_h, a_h) - Q_h^*(\tau_h, a_h)$$
$$= \widehat{r}_h(\tau_h, a_h) + [\widehat{\mathcal{P}}_h V_{h+1}](\tau_h, a_h) - r_h(\tau_h, a_h) - [\mathcal{P}_h V_{h+1}^*](\tau_h, a_h) + b_h^k(\tau_h, a_h)$$
$$\geq \underbrace{\left([\widehat{\mathcal{P}}_h - \mathcal{P}_h] V_{h+1}^*\right)(\tau_h, a_h)}_{(A)} + \underbrace{\widehat{r}_h(\tau_h, a_h) - r_h(\tau_h, a_h)}_{(B)} + b_h^k(\tau_h, a_h).$$

We lower bound $(A)$ and $(B)$ separately. For term $(A)$, we have

$$(A) = \sum_{\tau_{h+1}} V_{h+1}^*(\tau_{h+1}) \left(\widehat{p}_h(\tau_{h+1}|\tau_h, a_h) - p_h(\tau_{h+1}|\tau_h, a_h)\right)$$
$$= \sum_{\tau_{h+1}} V_{h+1}^*(\tau_{h+1}) \left(\widehat{p}_h(\tau_{h+1}|\tau_h, a_h) - p_h(\tau_{h+1}|\tau_h, a_h)\right) \mathbb{1}\{t_{h+1} = h + 1\}$$
$$+ \sum_{\tau_{h+1}} V_{h+1}^*(\tau_{h+1}) \left(\widehat{p}_h(\tau_{h+1}|\tau_h, a_h) - p_h(\tau_{h+1}|\tau_h, a_h)\right) \mathbb{1}\{t_{h+1} = t_h\}$$
$$= \underbrace{\sum_{s_{h+1}} V_{h+1}^*(\tau_{h+1}) \left((1 - \widehat{\lambda}_h)\widehat{p}_h(s_{h+1}|s_{t_h}, \mathbf{a}_{t_h:h}) - (1 - \lambda_h)p_h(s_{h+1}|s_{t_h}, \mathbf{a}_{t_h:h})\right)}_{(A_1)}$$
$$+ \underbrace{V_{h+1}^*(\{s_{t_h}, \mathbf{a}_{t_h:h}\})(\widehat{\lambda}_h - \lambda_h)}_{(A_2)}.$$

In $(A_1)$, $\tau_{h+1}$ is $\{s_{h+1}\}$. We bound $(A_1)$ as

$$(A_1) = \sum_{s_{h+1}} V_{h+1}^*(\tau_{h+1}) \Big((1 - \widehat{\lambda}_h)\widehat{p}_h(s_{h+1}|s_{t_h}, \mathbf{a}_{t_h:h}) - (1 - \lambda_h)\widehat{p}_h(s_{h+1}|s_{t_h}, \mathbf{a}_{t_h:h})$$
$$+ (1 - \lambda_h)\widehat{p}_h(s_{h+1}|s_{t_h}, \mathbf{a}_{t_h:h}) - (1 - \lambda_h)p_h(s_{h+1}|s_{t_h}, \mathbf{a}_{t_h:h})\Big)$$
$$= \sum_{s_{h+1}} V_{h+1}^*(\tau_{h+1})(1 - \lambda_h)\left(\widehat{p}_h(s_{h+1}|s_{t_h}, \mathbf{a}_{t_h:h}) - p_h(s_{h+1}|s_{t_h}, \mathbf{a}_{t_h:h})\right)$$
$$+ \sum_{s_{h+1}} V_{h+1}^*(\tau_{h+1})(\lambda_h - \widehat{\lambda}_h)\widehat{p}_h(s_{h+1}|s_{t_h}, \mathbf{a}_{t_h:h})$$
$$\overset{(i)}{\geq} -c_A H \sqrt{\frac{H\iota}{N_h(\tau_h, a_h)}} - H\left|\widehat{\lambda}_h - \lambda_h\right|,$$

where inequality $(i)$ invokes Hoeffding's inequality and holds with probability $1 - \gamma$ for any $\tau_h, a_h$ and some constant $c_A$. Term $(A_2)$ is immediately bounded by

$$(A_2) \geq -H\left|\widehat{\lambda}_h - \lambda_h\right|.$$

Putting $(A_1)$ and $(A_2)$ together, we derive

$$(A) \geq -c_A H \sqrt{\frac{H\iota}{N_h(\tau_h, a_h)}} - 2H\left|\widehat{\lambda}_h - \lambda_h\right|$$

with high probabilty. For term $(B)$, we have

$$(B) = \sum_{s_h} r(s_h, a_h)\left(\widehat{\mathfrak{b}}_h(s_h|\tau_h) - \mathfrak{b}_h(s_h|\tau_h)\right) \geq -c_B \sqrt{\frac{H\iota}{N_h(\tau_h, a_h)}}.$$

Taking $c = c_A + c_B$ and summing up $(A)$ and $(B)$, we have

$$Q_h(\tau_h, a_h) - Q_h^*(\tau_h, a_h) \geq -cH\sqrt{\frac{H\iota}{N_h(\tau_h, a_h)}} - 2H\left|\widehat{\lambda}_h - \lambda_h\right| + b_h^k(\tau_h, a_h).$$

We estimate $\lambda_h$ by its empirical average. In episode $k \geq 1$, we have access to $k$ i.i.d. realizations of Bernoulli random variable with rate $\lambda_h$ (observable or not). Therefore, by Hoeffding's inequality, we have

$$\left|\widehat{\lambda}_h^k - \lambda_h\right| \leq 2\sqrt{\frac{\log\frac{HK}{\gamma}}{k}} \leq 2\sqrt{\frac{\iota}{k}}.$$

Substituting into $Q_h^k(\tau_h, a_h) - Q_h^*(\tau_h, a_h)$ and reloading constant $c$ give rise to

$$Q_h^k(\tau_h, a_h) - Q_h^*(\tau_h, a_h) \geq -cH\left(\sqrt{\frac{H\iota}{N_h^k(\tau_h, a_h)}} + \sqrt{\frac{\iota}{k}}\right) + b_h^k(\tau_h, a_h) \geq 0.$$

The proof is complete. $\qquad\square$

**Regret analysis** We omit subscript "aug" to ease the presentation. The same derivation in the proof of Theorem 4.1 gives rise to

$$\left(Q_h^* - Q_h^{\pi_k}\right)(\tau_h^k, a_h^k) \leq \left(Q_h^k - Q_h^{\pi_k}\right)(\tau_h^k, a_h^k)$$
$$\leq \underbrace{\left([\widehat{\mathcal{P}}_h^k - \mathcal{P}_h][V_{h+1}^k - V_{h+1}^*]\right)(\tau_h^k, a_h^k)}_{(A)} + \left(\mathcal{P}_h[V_{h+1}^k - V_{h+1}^{\pi_k}]\right)(\tau_h^k, a_h^k) + 2b_h^k(\tau_h^k, a_h^k). \quad \text{(C.1)}$$

Lemma C.1 shows that $(A)$ can be written as

$$(A) = \sum_{s_{h+1}} [V_{h+1}^k - V_{h+1}^*](\tau_{h+1})(1 - \lambda_h)\left(\widehat{p}_h^k(s_{h+1}|s_{t_h}^k, \mathbf{a}_{t_h:h}^k) - p_h(s_{h+1}|s_{t_h}^k, \mathbf{a}_{t_h:h}^k)\right)$$
$$+ \sum_{s_{h+1}} [V_{h+1}^k - V_{h+1}^*](\tau_{h+1})(\lambda_h - \widehat{\lambda}_h^k)\widehat{p}_h^k(s_{h+1}|s_{t_h}^k, \mathbf{a}_{t_h:h}^k)$$
$$\leq \sum_{s_{h+1}} [V_{h+1}^k - V_{h+1}^*](\tau_{h+1})(1 - \lambda_h)\left(\widehat{p}_h^k(s_{h+1}|s_{t_h}^k, \mathbf{a}_{t_h:h}^k) - p_h(s_{h+1}|s_{t_h}^k, \mathbf{a}_{t_h:h}^k)\right) + H\left|\widehat{\lambda}_h^k - \lambda_h\right|$$
$$\leq (1 - \lambda_h)\sum_{s_{h+1}} [V_{h+1}^k - V_{h+1}^*](\tau_{h+1})\left(\widehat{p}_h^k(s_{h+1}|s_{t_h}^k, \mathbf{a}_{t_h:h}^k) - p_h(s_{h+1}|s_{t_h}^k, \mathbf{a}_{t_h:h}^k)\right) + 2H\sqrt{\frac{\iota}{k}}.$$

Following the derivation in (B.4), (B.5) and (B.6), we have

$$\texttt{Regret}(K) \leq e\sum_{k=1}^K \sum_{h=1}^H \left(\xi_h^k + \zeta_h^k + 2b_h^k + 2H\sqrt{\frac{\iota}{k}}\right)$$
$$\leq e\sum_{k=1}^K \sum_{h=1}^H \left(\xi_h^k + \zeta_h^k + 2b_h^k\right) + 2\sqrt{H^4 K\iota}.$$

where $\xi_h^k = \left(\mathcal{P}_h\left[V_h^k - V_h^{\pi_k}\right]\right)(\tau_h^k, a_h^k) - \left[V_{h+1}^k - V_{h+1}^{\pi_k}\right](\tau_{h+1}^k)$ is the martingale difference and $\zeta_h^k = c'\frac{SH^2\iota}{N_h^k(\tau_h^k, a_h^k)}$.

**Counting number summation** The summation over $\xi_h^k$ is standard. Using the Azuma-Hoeffding's inequality, we have

$$\sum_{k=1}^K \sum_{h=1}^H \xi_h^k \leq c_\xi\sqrt{KH^4\iota}.$$

It remains to find the summations involving $N_h^k(\tau_h^k, a_h^k)$. First, we show that the event $\mathcal{E}_m = \{h - t_h - 1 \leq m\}$, i.e., the maximal consecutive delay is upper bounded by $m > 0$, holds with high probability. We have

$$\mathbb{P}(\mathcal{E}_m) \leq \left(1 - H(1 - \lambda_0)^{m+1}\right)^K,$$

since $\lambda_0$ is a uniform lower bound of $\lambda_h$. Next, we provide an upper bound on $N_h^K(\tau_h, a_h)$. For a given tuple $(h, \tau_h, a_h, t_h)$, the consecutive missing length is $h - t_h - 1$. Such a missing pattern appears with probability at most $(1 - \lambda_0)^{h-t_h-1}$. As a consequence, denote $C_{h-t_h-1}^K$ as the number of $h - t_h - 1$ consecutive missings in $K$ episodes. With probability $1 - \gamma$, we have

$$C_{h-t_h-1}^K \leq K(1 - \lambda_0)^{h-t_h-1} + \sqrt{K(1 - \lambda_0)^{h-t_h-1}H\iota} + \iota.$$

by Bernstein's inequality in Lemma D.1. Furthermore, at a fixed time $h$, we use Lemma C.3 to bound the gap between two consecutive appearances of the same missing pattern. We instantiate Lemma C.3 with $\theta = (1 - \lambda_0)^{h-t_h-1}$ and obtain that the gap is bounded by $\left\lceil \frac{\iota}{-\log(1-(1-\lambda_0)^{h-t_h-1})} \right\rceil$ with probability $1 - \gamma$. Within the gap, the number of consecutive delays of length larger than $h - t_h - 1$ is bounded by

$$
\begin{aligned}
C_{\geq h-t_h-1} &\overset{(i)}{\leq} \left\lceil \frac{\iota}{-\log(1-(1-\lambda_0)^{h-t_h-1})} \right\rceil (1 - \lambda_0)^{h-t_h} \\
&\quad + \sqrt{\left\lceil \frac{\iota}{-\log(1-(1-\lambda_0)^{h-t_h-1})} \right\rceil (1 - \lambda_0)^{h-t_h}H\iota} + \iota \\
&\overset{(ii)}{\leq} \sqrt{2(1-\lambda_0)H\iota} + 2(1 - \lambda_0) + \iota,
\end{aligned}
$$

where inequality $(i)$ follows from Bernstein's inequality again and inequality $(ii)$ invokes the fact $x + \log(1 - x) \leq 0$ for $x \in [0, 1)$ and bounds $\lceil x \rceil$ by $x + 1$. Now we can bound the summation of the counting numbers. Conditioned on the event $\mathcal{E}_m$, we have

$$
\begin{aligned}
\sum_{k=1}^K \sum_{h=1}^H \sqrt{\frac{1}{N_h^k(\tau_h^k, a_h^k)}} &\overset{(i)}{\leq} \sum_{(h,\tau,a,t_h)} C_{\geq h-t_h-1} \sum_{i=1}^{N_h^K(\tau,a)} \sqrt{\frac{1}{i}} \\
&\leq 2\left(\sqrt{2(1-\lambda_0)H\iota} + 2(1-\lambda_0) + \iota\right) \sum_{(h,\tau,a,t_h)} \sqrt{N_h^K(\tau,a)} \\
&\overset{(ii)}{\leq} 2\left(\sqrt{2(1-\lambda_0)H\iota} + 2(1-\lambda_0) + \iota\right) \sum_{h,t_h} \sqrt{SA^{h-t_h}C_{h-t_h-1}^K} \\
&\leq 2\left(\sqrt{2(1-\lambda_0)H\iota} + 2(1-\lambda_0) + \iota\right) \\
&\quad \cdot \sum_{h,t_h} \sqrt{SA\left(K((1-\lambda_0)A)^{h-t_h-1} + \sqrt{K(A^2(1-\lambda_0))^{h-t_h-1}H\iota} + A^{h-t_h-1}\iota\right)} \\
&\overset{(iii)}{\leq} 2\left(\sqrt{2(1-\lambda_0)H\iota} + 2(1-\lambda_0) + \iota\right) \sum_{h,t_h} \sqrt{SA\left(K + \sqrt{KA^mH\iota} + A^m\iota\right)} \\
&\leq 2\left(\sqrt{2(1-\lambda_0)H\iota} + 2(1-\lambda_0) + \iota\right) H^2\sqrt{SA\left(K + \sqrt{KA^mH\iota} + A^m\iota\right)} \\
&\leq 2\sqrt{H^5 SA\iota^2 \left(K + \sqrt{KA^mH\iota} + A^m\iota\right)},
\end{aligned}
$$

where inequality $(i)$ follows since $N_h^k$ is repeated at most $C_{\geq h-t_h-1}$ times before getting an update and inequality $(ii)$ follows from Cauchy-Schwarz inequality, and inequality $(iii)$ invokes the assumption of $\lambda A \leq 1$. Moreover, conditioned on the event $\mathcal{E}_m$, we also have

$$
\sum_{k=1}^K \sum_{h=1}^H \frac{1}{N_h^k(\tau_h^k, a_h^k)} \leq \sum_{(h,\tau,a,t_h)} C_{\geq h-t_h-1} \sum_{i=1}^{N_h^K(\tau,a)} \frac{1}{i}
$$

$$\leq \left( \sqrt{2(1-\lambda_0)H\iota} + 2(1-\lambda_0) + \iota \right) \sum_{(h,\tau,a,t_h)} \log N_h^K(\tau, a)$$

$$\leq \iota H^{5/2} S A^{m+1} \log K.$$

**Combining the above**  On event $\mathcal{E}_m$, the regret is bounded by

$$\texttt{Regret}(K) \overset{(i)}{\leq} c \left( \sqrt{H^4 K \iota} + \sum_{k=1}^{K} \sum_{h=1}^{H} \left[ \frac{SH^2\iota}{N_h^k(\tau_h^k, a_h^k)} + H\sqrt{\frac{H\iota}{N_h^k(\tau_h^k, a_h^k)}} \right] \right)$$

$$\leq c \left( H^4 \sqrt{SA\iota^3 K \left( 1 + \sqrt{\frac{A^m H\iota}{K}} + \frac{A^m \iota}{K} \right)} + S^2 A^m \sqrt{H^9 \iota^6} + \sqrt{H^4 K \iota} \right),$$

where $c$ is a sufficiently large constant and we substitute the bonus functions into inequality $(i)$.

On the complement of $\mathcal{E}_m$, the regret is bounded by $H(1 - \mathbb{P}(\mathcal{E}_m)) \leq H^2 K(1-\lambda_0)^{m+1}$. We choose $m = \frac{1}{2} \left\lfloor \frac{\log K}{-\log(1-\lambda_0)} \right\rfloor$ such that $H(1 - \mathbb{P}(\mathcal{E}_m)) \leq H^2 K(1-\lambda_0)^{m+1} \leq H^2 \sqrt{K}$. We can now check that $A^{m+1} = \exp \left( \frac{\log A}{-\log(1-\lambda_0)} \log \sqrt{K} \right) \leq K^{\frac{1}{2(1+v)}}$. Therefore, combining the regret on event $\mathcal{E}_m$ and the complement event $\mathcal{E}_m^{\complement}$ leads to

$$\texttt{Regret}(K) \leq c \left( H^4 \sqrt{SAK\iota^3} + S^2 \sqrt{H^9 K^{\frac{1}{(1+v)}} \iota^6} \right).$$

The proof is complete. $\qquad\square$

## C.3  Supporting lemmas

**Lemma C.2.** Suppose Assumption 2.2 holds. With probability $1 - \gamma$ for some failure probability $\gamma > 0$, we have

$$\sum_{k=1}^{K} \sum_{h=1}^{H} \frac{1}{\sqrt{N_h^k(s_h^k, a_h^k)}} \leq \left\lceil \frac{\log \frac{HK}{\gamma}}{-\log(1-\lambda_0^2)} \right\rceil \sqrt{SAKH}.$$

*Proof of Lemma C.2.*  For any time $h$, we denote $\mathcal{K}^{\text{eff}}(h)$ as the collection of episodes that the $h$-th and $(h+1)$-th step observations are available. It is clear that the cardinality of $\mathcal{K}^{\text{eff}}(h)$ is bounded by $K$ for any $h$. Within each $\mathcal{K}^{\text{eff}}(h)$, we would like to bound the gap between two observations. Thanks to Lemma C.3, the gap is bounded by $q$ with probability $1 - K(1-\lambda_0^2)^{q+1}$. We set $K(1-\lambda_0^2)^{q+1} = \gamma/H$, which implies $q = \left\lceil \frac{\log \frac{HK}{\gamma}}{-\log(1-\lambda_0^2)} \right\rceil$. Therefore, for any time step $h$, available observations are at most separated by $q$ episodes.

With these notations, we bound

$$\sum_{k=1}^{K} \sum_{h=1}^{H} \frac{1}{\sqrt{N_h^k(s_h^k, a_h^k)}} \overset{(i)}{\leq} \left\lceil \frac{\log \frac{HK}{\gamma}}{-\log(1-\lambda_0^2)} \right\rceil \sum_{h=1}^{H} \sum_{k \in \mathcal{K}^{\text{eff}}(h)} \frac{1}{\sqrt{N_h^k(s_h^k, a_h^k)}}$$

$$\overset{(ii)}{\leq} \left\lceil \frac{\log \frac{HK}{\gamma}}{-\log(1-\lambda_0^2)} \right\rceil \sum_{h=1}^{H} \sum_{k=1}^{K} \frac{1}{\sqrt{N_h^k(s_h^k, a_h^k)}}$$

$$\overset{(iii)}{\leq} 2 \left\lceil \frac{\log \frac{HK}{\gamma}}{-\log(1-\lambda_0^2)} \right\rceil \sqrt{SAHK},$$

where inequality $(i)$ follows since $N_h^k$ will only be updated when $h \in \mathcal{K}^{\text{eff}}(h)$ and then repeat at most $\left\lceil \frac{\log \frac{HK}{\gamma}}{-\log(1-\lambda_0^2)} \right\rceil$ times, inequality $(ii)$ invokes the cardinality bound of $\mathcal{K}^{\text{eff}}(h)$, and inequality $(iii)$ follows from the standard pigeon-hole principle. $\qquad\square$

**Lemma C.3.** Let $\{u_i\}_{i=1}^k$ be i.i.d. Bernoulli random variables. Suppose $\mathbb{P}(u_i = 1) = \theta$. Define the largest gap between $u_i$'s as

$$g(k) = \sup\{j - i : u_i = 0 \text{ and } u_j = 0 \text{ with } u_\ell = 1 \text{ for } \ell = i+1, \ldots, j-1\}.$$

Then for any integer $q > 0$, the following tail probability bound holds

$$\mathbb{P}(g(k) > q) \le k\theta^{q+1}.$$

*Proof of Lemma C.3.* We denote $I_{\text{neg}} = \{\ell_1, \ldots, \ell_m\}$ as the index set for $u_{\ell_i} = 0$ when $i = 1, \ldots, |I_{\text{neg}}|$. Let $v_j = \ell_{j+1} - \ell_j$, which is a geometric random variable with a success rate $\theta$. Note that the cardinality of $I_{\text{neg}}$ is at most $k$. Therefore, we have

$$
\begin{aligned}
\mathbb{P}(g(k) > q) &\le \mathbb{P}(\max_{j=1,\ldots,k} v_j > q) \\
&= 1 - \mathbb{P}\left(v_j \le q \text{ for } j = 1, \ldots, k\right) \\
&= 1 - \left(1 - \theta^{q+1}\right)^k \\
&\le k\theta^{q+1},
\end{aligned}
$$

where the last inequality follows from $1 - k\theta^{q+1} \le (1 - \theta^{q+1})^k$. $\qquad\square$

# D  Helper concentration inequalities

**Lemma D.1** (Bernstein's inequality). Let $x_1, \ldots, x_n$ be i.i.d. zero mean random variables. Suppose $|x_i| \le M$ for any $i = 1, \ldots, n$. Then for all positive $t$, it holds

$$\mathbb{P}\left(\sum_{i=1}^n x_i > t\right) \le \exp\left(-\frac{\frac{1}{2}t^2}{\sum_{i=1}^n \mathrm{Var}[x_i] + \frac{1}{3}Mt}\right).$$

In particular, given a failure probability $\gamma < 1$, it holds

$$\mathbb{P}\left(\sum_{i=1}^n x_i > \sqrt{\sum_{i=1}^n \mathrm{Var}[x_i] \log \frac{1}{\gamma}} + M \log \frac{1}{\gamma}\right) \le \gamma.$$

*Proof of Lemma D.1.* The proof of Bernstein's inequality is standard, see for example [Wainwright, 2019, Section 2.1]. Here we verify the second claim. Let $\exp\left(-\frac{\frac{1}{2}t^2}{\sum_{i=1}^n \mathrm{Var}[x_i] + \frac{1}{3}Mt}\right) \le \gamma$ hold true. We find a suitable $t$ by

$$
\begin{aligned}
&\exp\left(-\frac{\frac{1}{2}t^2}{\sum_{i=1}^n \mathrm{Var}[x_i] + \frac{1}{3}Mt}\right) \le \gamma \\
\iff\quad & \frac{\frac{1}{2}t^2}{\sum_{i=1}^n \mathrm{Var}[x_i] + \frac{1}{3}Mt} \ge \log \frac{1}{\gamma} \\
\iff\quad & t^2 - \frac{2}{3}tM \log \frac{1}{\gamma} \ge \sum_{i=1}^n \mathrm{Var}[x_i] \log \frac{1}{\gamma} \\
\iff\quad & t \ge \sqrt{\sum_{i=1}^n \mathrm{Var}[x_i] \log \frac{1}{\gamma} + \frac{1}{9}M^2 \log^2 \frac{1}{\gamma}} + \frac{1}{3}M \log \frac{1}{\gamma}.
\end{aligned}
$$

It is enough to choose $t = \sqrt{\sum_{i=1}^n \mathrm{Var}[x_i] \log \frac{1}{\gamma}} + M \log \frac{1}{\gamma}$. $\qquad\square$

**Lemma D.2** (Hoeffding's inequality). Let $x_1, \ldots, x_n$ be i.i.d. random variables. Suppose $a_i \le x_i \le b_i$ for any $i = 1, \ldots, n$. Then for all positive $t$, it holds

$$\mathbb{P}\left(\left|\sum_{i=1}^n x_i - \mathbb{E}\left[\sum_{i=1}^n x_i\right]\right| > t\right) \le 2\exp\left(-\frac{2t^2}{\sum_{i=1}^n (b_i - a_i)^2}\right).$$

In particular, given a failure probability $\gamma < 1$, it holds

$$\mathbb{P}\left(\frac{1}{n}\left|\sum_{i=1}^{n} x_i - \mathbb{E}\left[\sum_{i=1}^{n} x_i\right]\right| > \sqrt{\frac{\sum_{i=1}^{n}(b_i - a_i)^2 \log \frac{2}{\gamma}}{2n^2}}\right) \leq \gamma.$$

*Proof of Lemma D.2.* The proof is standard; see [Wainwright, 2019, Section 2.1]. $\square$

**Lemma D.3** (Azuma-Hoeffding's inequality). Let $x_1, \ldots, x_n$ be a martingale adapted to filtration $\mathcal{F}_1 \subset \cdots \subset \mathcal{F}_n$. Suppose $\mathbb{E}[x_i - \mathbb{E}[x_i]|\mathcal{F}_{i-1}] = 0$ and $|x_i - \mathbb{E}[x_i]| \leq c_i$. Then for all positive $t$, it holds

$$\mathbb{P}\left(\sum_{i=1}^{n} x_i - \mathbb{E}[x_i] > t\right) \leq \exp\left(-\frac{t^2}{2\sum_{i=1}^{n} c_i^2}\right).$$

In particular, given a failure probability $\gamma < 1$, it holds

$$\mathbb{P}\left(\sum_{i=1}^{n} x_i - \mathbb{E}[x_i] > \sqrt{2\sum_{i=1}^{n} c_i^2 \log \frac{1}{\gamma}}\right) \leq \gamma.$$

*Proof of Lemma D.3.* The proof is standard and applies Lemma D.2. $\square$

