151 technical details, please feel free to skip this section.

### 3.1 Augmented MDP with expected reward

153 In the remainder of this section, we focus on the delayed observation case and defer the missing case
154 to Section 5. Define $\tau_h = \{s_{t_h}, \mathbf{a}_{t_h:h-1}, \delta_{t_h}\}$ as the augmented state, where $\delta_{t_h} \in [0, \Delta_{t_h}]$ is the
155 delayed steps after observing $(s_{t_h}, r_{t_h})$. Let $\mathcal{S}_{\text{aug}}$ denote the augmented state space of all possible
156 $\tau$'s. Then the original MDP with delayed observations can be reformulated into a state-augmented
157 one $\text{MDP}_{\text{aug}} = (\mathcal{S}_{\text{aug}}, \mathcal{A}, H, R_{\text{aug}}, P_{\text{aug}})$. The reward is defined as

$$r_{h,\text{aug}}(\tau_h, a_h) = \mathbb{E}\left[r_h(s_h, a_h) | \tau_h, a_h\right],$$

158 which is the expected reward given the nearest past state $s_{t_h}$ and history actions $\mathbf{a}_{t_h:h}$. We can define
159 belief distribution $\mathfrak{b}_h(s | \tau_h) = \mathbb{P}(s_h = s | \tau_h)$. Then $r_{h,\text{aug}}(\tau_h, a_h) = \mathbb{E}_{s \sim \mathfrak{b}_h(\cdot | \tau_h)}[r(s, a_h)]$. Belief
160 distributions are widely adopted in partially observed MDPs [Ross et al., 2007, Poupart and Vlassis,
161 2008]. We will frequently use the belief distribution to study the expressivity of $\Pi_e$ in Section 4.2.

The transcription probabilities $P_{\mathrm{aug}}$ are sparse. For any $\tau_h = \{s_{t_h}, \mathbf{a}_{t_h:h-1}, \delta_{t_h}\}$ and $\tau_{h+1} = \{s_{t_{h+1}}, \mathbf{a}_{t_{h+1}:h}, \delta_{t_{h+1}}\}$, we have

| $p_{h,\mathrm{aug}}(\tau_{h+1}|\tau_h, a_h)$ | Condition |
|---|---|
| $\mathtt{M}_a(\tau_h, \tau_{h+1})\theta_{\mathrm{delay}}(s_{t_h}, a_{t_h}, \delta_{t_h})p_h(s_{t_{h+1}}|s_{t_h}, a_{t_h})$ | if $\delta_{t_{h+1}} = 0$ and $t_{h+1} = t_h + 1$ |
| $\mathtt{M}_a(\tau_h, \tau_{h+1})(1 - \theta_{\mathrm{delay}}(s_{t_h}, a_{t_h}, \delta_{t_h}))$ | if $\delta_{t_{h+1}} = \delta_{t_h} + 1$ and $t_{h+1} = t_h$ |
| $0$ | otherwise |

where $\mathtt{M}_a(\tau_h, \tau_{h+1})$ indicates whether the rolling actions are matched, i.e.,
$$\mathtt{M}_a(\tau_h, \tau_{h+1}) = \mathbb{1}\{\mathbf{a}_{t_h:h-1} = \mathbf{a}_{t_{h+1}:h-1}\},$$
and $\theta_{\mathrm{delay}}(s_{t_h}, a_{t_h}, \delta_{t_h})$ is defined as

$$\theta_{\mathrm{delay}}(s_{t_h}, a_{t_h}, \delta_{t_h}) = \mathbb{P}(\Delta_{t_h} = \delta_{t_h}|s_{t_h}, a_{t_h}, \delta_{t_h}) = \frac{\mathbb{P}(\Delta_{t_h} = \delta_{t_h}|s_{t_h}, a_{t_h})}{1 - \sum_{\delta < \delta_{t_h}} \mathbb{P}(\Delta_{t_h} = \delta|s_{t_h}, a_{t_h})}.$$

The factored form of $\theta_{\mathrm{delay}}(s_{t_h}, a_{t_h}, \delta_{t_h})p(s_{t_{h+1}}|s_{t_h}, a_{t_h})$ follows from the conditional independence in Assumption 2.1. We define $Q$-functions and value functions as follows. For any $\tau_h, a_h$ and policy $\pi \in \Pi_{\mathrm{e}}$, we have

$$Q_{h,\mathrm{aug}}^\pi(\tau_h, a_h) = \mathbb{E}^\pi\left[\sum_{h'=h}^H r_{h,\mathrm{aug}}(\tau_{h'}, a_{h'})\Big|\tau_h, a_h\right] \quad \text{and}$$

$$V_{h,\mathrm{aug}}^\pi(\tau_h) = \left\langle Q_{h,\mathrm{aug}}^\pi(\tau_h, \cdot), \pi_h(\cdot|\tau_h)\right\rangle.$$

We note that $V_h^\pi$ is equivalent to $V_{h,\mathrm{aug}}^\pi$ for the same executable policy $\pi \in \Pi_{\mathrm{e}}$. We also denote $\mathcal{P}_{h,\mathrm{aug}}$ as the transition operator corresponding to $P_{\mathrm{aug}}$. It can be checked that
$$Q_{h,\mathrm{aug}}^\pi(\tau_h, a_h) = r_{h,\mathrm{aug}}(\tau_h, a_h) + [\mathcal{P}_{h,\mathrm{aug}}V_{h,\mathrm{aug}}^\pi](\tau_h, a_h).$$
$\mathrm{MDP}_{\mathrm{aug}}$ also appears in makes all the policies in $\Pi_{\mathrm{e}}$ executable and Markov. Meanwhile, the reward function keeps track of all the expected reward for $H$ steps. Although the expanded state space $\mathcal{S}_{\mathrm{aug}}$ is much more complicated than the original state space $\mathcal{S}$, the sparse structures in the transition probabilities still allow an efficient exploration. We note that $p_{h,\mathrm{aug}}$ only depends on the delay distribution and one-step Markov transitions. However, there is still one caveat for learning in $\mathrm{MDP}_{\mathrm{aug}}$ – the reward function depends belief distributions, which involve multi-step transitions.

## 3.2 Augmented MDP with past reward

To tackle the aforementioned challenge, we further define $\widetilde{\mathrm{MDP}}_{\mathrm{aug}} = (\widetilde{\mathcal{S}}_{\mathrm{aug}}, \mathcal{A}, \widetilde{H}, \widetilde{R}_{\mathrm{aug}}, \widetilde{P}_{\mathrm{aug}})$ that shares the optimal policy in $\mathrm{MDP}_{\mathrm{