# OpenReview forum: "Efficient RL with Impaired Observability: Learning to Act with Delayed and Missing State Observations"
_NeurIPS.cc/2023/Conference — NeurIPS 2023 poster_

### Official Review · Reviewer_61zK · 2023-06-12

**Soundness:** 3 good
**Presentation:** 3 good
**Contribution:** 2 fair
**Rating:** 5
**Confidence:** 4

**Summary:**

This paper studies online learning in tabular MDPs with impaired observability, which means one of two models: states are observed in stochastic delay or some observations are missing (sampled independently). The authors present algorithms based on optimism and prove regret bounds that scale optimally with $S,A,K$ and polynomially with $H$.

**Strengths:**

**Summary:** I don't have a strong conviction about this paper. On the one hand the paper is well-written and the model is interesting and novel, but on the other hand the regret bounds are straightforward and there is no interesting algorithmic novelty in my opinion.

**Strengths:**
1. RL with impaired observability is an interesting model which obviously has a lot of real-world applications. The authors do a good job of introducing the model, the challenges and the motivation.
2. The paper is generally well-written and easy to follow (except for a few technical parts, see questions). It conveys well the need for the impaired observability model and the idea behind the algorithmic approaches to solve it.
3. The algorithms are simple and the regret bounds are close to optimal (although one can argue about what is close to optimal in this setting).

**Weaknesses:**

**Weaknesses:**
1. I think that the novelty of this paper is limited to the definition of the new models. In terms of algorithmic contribution, the application of optimism here is straightforward and so is the regret analysis. In fact, unless I am missing something, the augmented MDP is a factored MDP and then the algorithms (and analysis) actually exist already, so there is no surprise that the regret is polynomial. If that is the case, then the authors should also discuss the factored MDP literature, e.g., "Near-optimal Reinforcement Learning in Factored MDPs" by Ian Osband and Benjamin Van Roy.
2. The algorithms run in exponential time in $H$. I understand that this is probably inevitable, but it still makes the contributions of this paper very limited.
3. The regret analysis is similar to previous works and does not investigate in-depth the dependence on the delays or the missing observations rate. The regret of algorithm 2 "hides" the dependence in the distribution of the delays in the extra $H$ factors. I think that the actual dependence is interesting in this setting because it can show which delays actually become a problem. While the dependence on the missing rate is shown in the regret of algorithm 3, the optimality is not discussed and instead there is an additional algorithm that replaces this optimality with another assumption (that to me looks like it makes the whole thing easy because very few observations are missing). Moreover, There is not enough discussion about the assumptions 2.1 and 2.2, and what happens if they are relaxed.

**Questions:**

**Questions:**
1. The definition of the augmented MDP is very hard to follow. Can that authors please explain in words what this MDP looks like?
2. Can the authors please discuss the assumptions a little more? For example, is the assumption that feedback arrives in order necessary? what happens when there are large delays and why is the regret always bounded within $H$?
3. Is the augmented MDP a factored MDP?
4. Can you think of a scenario where the computational complexity is not exponential?

**Limitations:**

yes

---

> ### Author Rebuttal · Authors · 2023-08-10
>
> Thank you for your comments!
>
> >**Q1**: The novelty of this paper is limited to the definition of the new models. It appears to be a factored MDP with known solutions.
>
> **A1**: We disagree respectfully. Our augmented MDP is not a factored MDP. Factored MDPs encode rewards and transitions exhibiting some conditional independence structure among factors. Although our augmented MDP has sparse transition probabilities, it does not have immediate conditional independence structures. Thus, there was no prior solution to the delayed MDP except for naive exponential-regret methods.
>
> More importantly, we are first to provide concrete regret bounds scale with $\sqrt{S A}$ by developing several novel techniques regarding the augmented MDP formulation and bonus function construction and analysis.
>
> 1). To tackle the exponentially large state space in the augmented MDP, we construct a reformation tilde_MDP_aug with past reward and an extended horizon. This new reformulation provably attains the same expected reward as MDP_aug and serves as the basis of Algorithm 2.
>
> 2). In Algorithm 2, we provide a novel construction on the bonus function to ensure optimism with high probability. The intriguing fact of the bonus function is that it is akin to the bonus of UCBVI applied to the original MDP. However, this is a consequence of our analysis exploiting the sparse structure in the transition kernel. A direct application of the UCBVI algorithm to tilde_MDP_aug will easily end up with annoying $N(s_{t_h}, a_{t_h:h-1})$ counting numbers, which leads to exponential regret. In addition, we also provide uncertainty quantification on the arbitrary delay distribution, which to our best knowledge is never done in prior work.
>
> >**Q2**: The algorithms run in exponential time.
>
> **A2**: There seems to be some misunderstanding. While our main focus is the learning efficiency, our proposed algorithms actually work with any planning oracle and achieve polynomial sample complexity. The technical contributions are discussed in response A1. Depending on the planning oracle to be chosen by a practical user, the time complexity might vary and would be exponential in $H$ only with the worst choice. In practice, one often solves the planning problem using policy gradient methods with function approximation, which proved quite efficient even in large-scale problems.
>
> >**Q3**: The regret analysis is similar to previous works and does not investigate in-depth the dependence on the delays or missing observations rate.
>
> **A3**: We do not assume any distribution on the length of delay. Hence, the regret bound holds for arbitrary delay in the worst case. The idea is that we consider finite-horizon MDPs, therefore, we can truncate the length of delay at $H$ (Line 221-225).
>
> When the delay is much better behaved, e.g., is bounded by some constant $D_{\max}$ smaller than $H$, we can obtain better regret bounds. With the same algorithm, a very slight modification on our proof leads to a regret of $\tilde{\mathcal{O}}((D_{\max}+1)^{3/2} H^{5/2} \sqrt{SAK})$. As can be seen, when $D_{\max}$ is small, the regret becomes small. Moreover, when $D_{\max} = 0$, i.e., there is no delay, the regret recovers that in standard MDPs without delay.
>
> Please also see the response A3 to **Reviewer yWj8** for technical details of how to obtain the modified regret bounds.
>
> >**Q4**: While the dependence on the missing rate is shown in the regret of algorithm 3, the optimality is not discussed and instead there is an additional algorithm that replaces this optimality with another assumption.
>
> **A4**: Proposition 5.1 confirms that obtaining polynomial regret with missing observations is possible. Yet the $S^2$ dependence may not be optimal. With an assumption on the missing rate, we indeed show a $\sqrt{SA}$ regret, which matches the minimax optimal dependence on S, A in the standard MDP setting. We further discuss the assumption on the missing rate in Line 304 - 310, while leaving the study of missing rates larger than $1/A$ as future work.
>
> The analysis to Theorem 5.2 is rather complicated, in contrast to the seemingly “easy” small missing rate. The major difficulty is to handle the summation over multi-step counting numbers (from Line 593 to the end of Appendix C).
>
> >**Q5**: Moreover, there is not enough discussion about the assumptions 2.1 and 2.2, and what happens if they are relaxed.
>
> **A5**: Assumption 2.1 and 2.2 are fairly general themselves and encode arbitrary delay and missing distributions (see a discussion in Line 123-127). Our theory holds even in the worst scenario.
>
> To further relax Assumption 2.1 and 2.2, such as allowing interarrival time $\Delta_h$ to be negative, i.e., $s_{h}$ can be observed before $s_{h-1}$, and missing rate dependent on the underlying state, goes beyond the scope of the current paper.
>
> >**Q6**: The definition of the augmented MDP is very hard to follow.
>
> **A6**: Roughly speaking, the augmented MDP is to form an enlarged state space with the nearest observed state and all the history actions thereafter (see definition of $\tau_h$ in Line 154). The reward and transition probabilities are slightly complicated, but built upon the original MDP. The reward function is the expected reward (Line 157 - 158) over the belief of the unseen current state, given all the observed information. The transition probabilities are sparse, as past actions cannot be altered (Line 164 - 165).
>
> >**Q7**: A scenario where the computational complexity is not exponential?
>
> **A7**: While investigating practical planning algorithms for specific problem is beyond the scope of our paper, we believe that our augmented MDP formation is compatible with *any* planning oracle for accelerated solution of RL.
>
> As a particular example, when the underlying transition is nearly deterministic, Thomas J Walsh, Ali Nouri, Lihong Li, and Michael L Littman. “Planning and learning in environments with delayed feedback” show a polynomial planning complexity.

---

> > ### Comment · Reviewer_61zK · 2023-08-13
> >
> > I thank the reviewers for all their detailed responses. I will keep my positive score.

---

### Official Review · Reviewer_Mu3C · 2023-07-06

**Soundness:** 3 good
**Presentation:** 2 fair
**Contribution:** 3 good
**Rating:** 6
**Confidence:** 3

**Summary:**

The paper considers reinforcement learning with delayed and missing observations. It is shown that a MDP with delayed observations is equivalent to an augmented MDP with perfect observations. Based on the augmented MDP, an optimistic algorithm is proposed for delayed MDPs and it is shown to achieve a near-optimal regret order. For MDPs with missing observations, two optimistic algorithms are proposed with near-optimal regret order under some assumptions on the missing rate.

**Strengths:**

- For MDPs with delayed observations, under some independent assumption and an assumption requiring the delayed observations will still arrive in order, the paper constructs an equivalent augmented MDP with perfect observations. Based on the augmented MDP, an optimistic algorithm is proposed for the delayed MDP. Despite the increased cardinality of the augmented MDP, the proposed algorithm is shown to achieve a regret with sharp dependence on the state and action spaces.

- The effect of observation delay on performance degradation is discussed with some performance gap bounds provided between the optimal policy with and without observation delays.

- For MDPs with missing observations, one optimistic algorithm is proposed and its regret is shown to be sub-linear in the number of episodes.

- When the missing rate is small, another algorithm based on the augmented MDP is proposed. This algorithm is shown to have near-optimal regret order if the number of episode is large enough.

**Weaknesses:**

- There are some possible typos in key equations which may highly affect the derivations and results.
  - In the definition of the transition below line 164, the collections of actions for time $h$ and $h+1$ may have different sizes. Is that just some typo or some additional things need to be handled?
  - In the equation above line 182, should the reward be $r_{t_h}$?
  - In the bonus function in Algorithm 2, $a_h$ seems to be a typo.

- There seems to be several missing steps in the proof of Proposition 5.1. The entire derivation of the proof should be conditioned on some high probability event for the inequalities to hold, but those steps are missing. Inequality (i) is claimed to be true from valid optimism, but optimism for Algorithm 3 is not provided. Inequality (ii) might also need some further discussion.

**Questions:**

- Can the authors fix some of the typos and missing steps?

**Limitations:**

Limitations are adequately addressed in the paper.

---

> ### Author Rebuttal · Authors · 2023-08-10
>
> Thank you for your comments!
>
> >**Q1**: In the definition of the transition below line 164, the collections of actions for time $h$ and $h+1$ may have different sizes. Is that just some typo or some additional things need to be handled?
>
> **A1**: Varying sizes of actions is a consequence of our general delay distribution, i.e., at time $h$, there is a possibility that there is no new state observation. We can only increase the action sequence in the augmented space. Yet our analysis tackles the varying sizes of actions. We bound the worst case cardinality of the augmented state space in Line 467 and the optimism holds for any augmented state.
>
> >**Q2**: In the equation above line 182, should the reward be $r_{t_h}$?
>
> **A2**: Thanks for pointing out the typo.
>
> >**Q3**: In the bonus function in algorithm 2, $a_h$ seems to be a typo.
>
> **A3**: Thanks for pointing out the typo.
>
> >**Q4**: There seems to be several missing steps in the proof of Proposition 5.1. The entire derivation of the proof should be conditioned on some high probability event for the inequalities to hold, but those steps are missing. Inequality (i) is claimed to be true from valid optimism, but optimism for Algorithm 3 is not provided. Inequality (ii) might also need some further discussion.
>
> **A4**: We are confident about the correctness and rigor of our proof. Thanks for the comment and we will add more clarifications.
>
> The high probability event we are conditioned on is that the ground-truth transition (denoted by $\theta^*$ notation) falls into the set $\mathcal{B}_k$. This can be shown by a direct application of Hoeffding’s inequality. We will add this argument in the next version.
>
> Inequality (i) of Proposition 5.1 follows from Line 4 of Algorithm 3. As the ground truth environment is contained in $\mathcal{B}_k$, the value of the best executable policy for the ground truth environment is no larger than the largest value under the double maximization.
>
> Inequality (ii) follows from a standard telescoping expansion of the value function over time by utilizing the fact that $Q_h(s, a) = r_h(s, a) +[P_h V_{h+1}](s, a)$ for any $h$. We apply this expansion to both $V_{\theta^k}^{\pi^k}$ and $V_{\theta^*}^{\pi^k}$.

---

> ### Comment · Area_Chair_kJEV · 2023-08-16
> **Are you satisfied by the answers?**
>
> Dear reviewer,
>
> Would you please indicate whether the authors' response is satisfactory for you? If not, please engage with the authors, so we can get a better assessment of this work.
>
> Thank you,
> Area Chair

---

> > ### Comment · Area_Chair_kJEV · 2023-08-18
> > **Follow-up**
> >
> > I'll like to follow up on this. You raised an issue with the proof of Prop 5.1. Is the authors' response satisfactory?

---

> > > ### Comment · Reviewer_Mu3C · 2023-08-19
> > >
> > > I appreciate the authors' response. The steps of proving Proposition 5.1 sound reasonable, but I don't think we can tell their correctness without having the complete proof. Moreover, the value of the constant c is in Algorithm 3 is not available.
> > >
> > > There is another issue with Proposition 5.1. When $\lambda_0=1$, the first term in the bound becomes zero. The remaining regret then has a log-dependence of S which does not seem to match the standard MDP case as the author responded to another reviewer. Is there something wrong or did I miss something?

---

> > > > ### Author Response · Authors · 2023-08-19
> > > >
> > > > Dear Reviewer,
> > > >
> > > > Thank you for your follow up questions.
> > > >
> > > > > **Q1**: The steps of proving Proposition 5.1 sound reasonable, but I don't think we can tell their correctness without having the complete proof. Moreover, the value of the constant c is in Algorithm 3 is not available.
> > > >
> > > > **A1**: We would like to confirm the correctness of Proposition 5.1. The proof details described in the previous response are all standard. In particular, in the $k$-th episode, we denote the event $\mathcal{E}_k = (p_h^{\theta^*}(\cdot | s, a) \in \mathcal{B}_k)$, which refers to the ground-truth transition belonging to the confidence set. This event holds with high probability as
> > > > $$|| \hat{p}_h(\cdot | s, a) - p_h^{\theta^*}(\cdot | s, a) ||_1 \geq t $$
> > > > with probability
> > > > $$(2^S - 2) \exp(-N^k(s, a)t^2 / 2).$$
> > > > (See equation (44) in [Jaksch et al., 2010] as mentioned in Line 266 and also [1].) Taking $t = 4 \sqrt{\frac{S \iota}{N^k(s, a)}}$ leads to the confidence set $\mathcal{B}_k$ in Algorithm 3 with constant $c = 4$.
> > > >
> > > > Conditioned on event $\mathcal{E}_k$, we have $p_h^{\theta^*} \in \mathcal{B}_k$, which leads to the valid optimism in inequality (i) of the proof of Proposition 5.1. Inequality (ii) is the standard performance difference expansion, as explained by the telescoping sum.
> > > >
> > > > [1] "Inequalities for the L1 Deviation of the Empirical Distribution", Tsachy Weissman, Erik Ordentlich, Gadiel Seroussi, Sergio Verdu, Marcelo J. Weinberger
> > > >
> > > > > **Q2**: There is another issue with Proposition 5.1. When $\lambda_0 = 1$, the first term in the bound becomes zero. The remaining regret then has a log-dependence of $S$ which does not seem to match the standard MDP case as the author responded to another reviewer. Is there something wrong or did I miss something?
> > > >
> > > > **A2**: Please refer to the proof of Proposition 5.1 in Line 557 - 558, where the dependence on $\lambda_0$ is given as $\lceil \frac{1}{-\log (1-\lambda_0^2)}\rceil$ with the ceiling function. As $\lambda_0$ approaching $1$ (strictly speaking, we cannot directly substitute $\lambda_0 = 1$ in Proposition 5.1), this term leads to a value of $1$ but not vanishing. Therefore, Proposition 5.1 always has a $S^2$ dependence and matches the standard MDP case with no issue. We will add the ceiling function to the statement of Proposition 5.1.
> > > >
> > > > We hope these clarifications helpful. Let us know if there is anything still lingering.

---

> > > > ### Author Response · Authors · 2023-08-21
> > > >
> > > > Dear Reviewer Mu3C,
> > > >
> > > > Thank you again for taking the time to review and discuss our paper! We have provided response to your correctness questions about Proposition 5.1. Could you please let us know if our response resolved your concerns? As the author-reviewer discussion ends today, please don’t hesitate to contact us if you have any further questions.
> > > >
> > > > Best,
> > > > Authors

---

> > > > > ### Comment · Reviewer_Mu3C · 2023-08-21
> > > > >
> > > > > Thanks the authors for the further clarifications.

---

### Official Review · Reviewer_kTAd · 2023-07-07

**Soundness:** 3 good
**Presentation:** 2 fair
**Contribution:** 3 good
**Rating:** 5
**Confidence:** 3

**Summary:**

This paper aims to provide a theoretical analysis of reinforcement learning with delayed and missing state observations, and establish near-optimal regret bounds, for RL in both the delayed and missing observation settings. Despite impaired observability posing significant challenges to the policy class and planning, the results demonstrate that learning remains efficient, with the regret bound optimally depending on the state-action size of the original system.  The policy under impaired observability is evaluated.

**Strengths:**

1. The problem addressed is critical for applying RL to real-world scenarios.
2. The theoretical analysis is comprehensive and well-executed.

**Weaknesses:**

1. The paper does not include preliminary experimental results, which may hinder the empirical understanding of the methods.
2. Building connections between the theoretical analysis and existing empirical studies in the literature would be beneficial.

**Questions:**

1. How can the theoretical analysis inspire further algorithm design to address impaired observations?
2. What are the main challenges preventing you from conducting empirical studies for the proposed method?

**Limitations:**

1. The scalability for large action spaces is limited.
2. The lack of empirical studies makes the applicability of the method uncertain.

---

> ### Author Rebuttal · Authors · 2023-08-10
>
> Thank you for your comments!
>
> >**Q1**: The paper does not include preliminary experimental results, which may hinder the empirical understanding of the methods.
>
> **A1**: Thanks for the suggestion. In the RL theory literature, developing theory for tabular MDP is usually the first step, and most papers in this category did not need experiments due to the nature of tabular problems [1-3].
>
> In contrast, we followed your suggestion and **conducted additional experiment on a toy model (see https://openreview.net/attachment?id=Pd2GMnkFUx&name=pdf for results)**. In particular, we consider constant delays and set the MDP with horizon $H = 6$. Detailed transitions and reward are summarized in Table 1 and Table 2  in the attached PDF. We vary the length of delay to be 1 and 2, and run our proposed policy learning Algorithm 2 for 10000 episodes. The regret is plotted in Figure 1. As can be seen, the regret increases as the length of delay increases. In Figure 2, we investigate the performance degradation caused by delayed observations. When the transition is almost deterministic, we observe that the gap is relatively small. However, when the transition is more random, the gap increases. These observations validate our main theory results such as Proposition B.2 and 4.2.
>
> [1] “Near-optimal reinforcement learning in polynomial time” by Michael Kearns and Satinder Singh.
>
> [2] “Minimax Regret Bounds for Reinforcement Learning” by Mohammad Gheshlaghi Azar, Ian Osband, Rémi Munos.
>
> [3] “Near-optimal regret bounds for reinforcement learning” by Thomas Jaksch, Ronald Ortner, and Peter Auer.
>
> >**Q2**: The scalability for large action spaces is limited.
>
> **A2**: Our main focus is the learning efficiency. The proposed algorithms actually work with any planning oracle and achieve polynomial sample complexity. Depending on the planning oracle to be chosen by a practical user, the time complexity might vary and would be exponential in only with the worst choice. In practice, one often solves the planning problem using policy gradient methods with function approximation, which proved quite efficient even in large-scale problems.

---

> ### Comment · Area_Chair_kJEV · 2023-08-16
> **Are you satisfied by the answers?**
>
> Dear reviewer,
>
> Would you please indicate whether the authors' response is satisfactory for you? If not, please engage with the authors, so we can get a better assessment of this work.
>
> Thank you,
> Area Chair

---

### Official Review · Reviewer_yWj8 · 2023-07-08

**Soundness:** 3 good
**Presentation:** 2 fair
**Contribution:** 3 good
**Rating:** 7
**Confidence:** 3

**Summary:**

This paper studies reinforcement learning in the setting where MDP is with delayed state information, but delayed observations still arrive in order, and in the setting where the state information of MDPs could be missed and never be observed. For both settings, this paper provides provably efficient RL algorithms with sub-linear regrets, whose dependencies on the number of episodes are optimal. In the setting with delayed state information, this paper shows the impact of delay $d_h$ on the performance of the optimal policy. In the setting with missing state information, this paper shows the impact of observation rate $\lambda_h$ on the regret. The results seem interesting to me.

**Strengths:**

1. This paper studies MDPs with delayed state information and MDPs with missing state information, which seems to be an important problem.

2. This paper provides algorithms for these two settings, and proves the regret guarantees.

3. This paper shows the impacts of the values of delays and observation rates on the performance of the policies for learning in MDPs with impaired observability.

**Weaknesses:**

1. The novelty of the proposed algorithms is not clear to me. For example, algorithm 2 seems to be a simple application of the tabular RL algorithm to the case with delayed state information?

2. It is not clear for me how to understand the regret. For example, why the benchmark in the regret is the optimal policy with impaired observability? Is it more reasonable to compare with stronger optimal policy, e.g., one with full observability? Since if the optimal policy is also with impaired observability, why is the problem more challenging?

**Questions:**

1. Why is the regret in Theorem 4.1 for delayed state information independent of delay? Is it because the optimal policy also suffers from the delay? Intuitively, the performance of the online policy with larger delay should be worse. For example, should we hope much better regret in the case with no delay than the case with infinite delay?

2. Even though when the delay $d_h$ is 0, there is still a gap in the regret in Theorem 4.1? Could you give some insights whether this can actually be sharpened or why is this true?

3. Similarly, in section 5 with missing state information, even when the missing rate is 0, the regret seems to keep to be larger than the best regret that could be obtained?

4. Above line 119, the executable policy class is defined based on the action sequence. However, based on the notation, the action is always known, why the action sequence $a_{t_h:h-1}$ is important there?

**Limitations:**

Please see weakness and questions above.

---

> ### Author Rebuttal · Authors · 2023-08-10
>
> Thank you for your comments!
>
> >**Q1**: Novelty of the proposed algorithms.
>
> **A1**: A naive application of tabular RL algorithm to our problem would yield exponential $\tilde{\mathcal{O}}({\rm poly}(H) \sqrt{SA^HK})$ regret. We are able to achieve regret that scales with $\sqrt{S A}$ by utilizing structures of the delayed MDP and novel construction on bonus functions.
>
> 1). To tackle the exponentially large state space in the augmented MDP, we construct a reformation tilde_MDP_aug with past reward and an extended horizon. This new reformulation provably attains the same expected reward as MDP_aug and serves as the basis of Algorithm 2.
>
> 2). In Algorithm 2, we provide a novel construction on the bonus function to ensure optimism with high probability. The intriguing fact of the bonus function is that it is akin to the bonus of UCBVI applied to the original MDP. However, this is a consequence of our analysis exploiting the sparse structure in the transition. A direct application of the UCBVI algorithm to tilde_MDP_aug will easily end up with annoying $N(s_{t_h}, a_{t_h:h-1})$ counting numbers, which leads to exponential regret. In addition, we also provide uncertainty quantification on the arbitrary delay distribution, which to our knowledge is never done in prior work.
>
> >**Q2**: How to understand the regret.
>
> **A2**: Our theory provides comparisons to both the best executable policies and the even stronger full-observability optimal policies, where the former is termed as regret (Line 204) and the latter is termed as gap (Line 232-233). Briefly speaking, regret measures the learnability of the best executable policy and the gap characterizes the unavoidable performance degradation due to impaired observability.
>
> In our separation theory (Proposition 4.2), we have shown that the gap is heavily instance-dependent and can be large in the worst case.  We also provided a general bound on the performance gap in Proposition B.2, which is deferred to appendix due to space limit. In the worst case, there is a constant gap compared to the optimal policy without impaired observability, indicating the divergence of the gap as $K$ increases.
>
> On the other hand, even compared to the optimal executable policy, as mentioned in the introduction, naive approaches easily result in an exponential regret. Our algorithm and analysis led to the first **tractable** solution to RL with impaired observability, i.e., achieving complexity that scales polynomially with $S$ and $A$.
>
> >**Q3**: Regret in Theorem 4.1 independent of delay?
>
> **A3**: We do not assume any distribution on the length of delay. Hence, the regret bound holds for arbitrary delay in the worst case. The idea is that we consider finite-horizon MDPs, therefore, we can truncate the length of delay at $H$ (Line 221-225).
>
> When the delay is much better behaved, e.g., is bounded by some constant $D_{\max}$ smaller than $H$, we can obtain better regret bounds. With the same algorithm, a very slight modification on our proof leads to a regret of $\tilde{\mathcal{O}}((D_{\max}+1)^{3/2} H^{5/2} \sqrt{SAK})$. As can be seen, when $D_{\max}$ is small, the regret becomes small. Moreover, when $D_{\max} = 0$, i.e., there is no delay, the regret recovers that in standard MDPs without delay. (Technical details of how to obtain these regret bounds are provided at the end.) It is worth pointing out that the dependence on $S$, $A$, and $K$ stays the same and is sharp.
>
> Technical details on how to show the $\tilde{\mathcal{O}}((D_{\max}+1)^{3/2} H^{5/2} \sqrt{SAK})$ regret.
>
> 1) The cardinality of S_aug is bounded by $HSA^{D_{\max}+1}$ now (Line 467 in Appendix B.1).
>
> 2) The high probability event holds uniformly over S_aug. Therefore, the confidence band is narrowed by replacing $H$ with $D_{\max} + 1$ in the square root.
>
> 3) In Eqn. (B.7) and (B.10), we replace a factor of $H$ by $D_{\max}+1$ as the delay is bounded by $D_{\max}$.
>
> Putting 1), 2), and 3) together leads to the claimed regret bound.
>
> 4) In the case of no delay, we do not need to estimate the distribution of delays. Therefore, the bonus function is simplified and there is no need to sum up in Eqn. (B.4). With these simplifications, we recover the regret bound in standard MDPs.
>
> >**Q4**: When the delay $d_h$ is 0. There is still a gap in the regret in Theorem 4.1?
>
> **A4**: Our regret in Theorem 4.1 is sharp in S, A, and K. As mentioned, we recover the minimax optimal dependence on S, A, and K in the standard MDP setting. Our analysis covers arbitrary delay in the worst case. When the length of delay is bounded, we can modify the regret bound as shown in the previous response A3.
>
> >**Q5**: When the missing rate is 0, the regret seems to be larger than the best regret?
>
> **A5**: In Proposition 5.1 (the last inequality above Line 558), when the observation rate $\lambda_0$ is 1, i.e., no missing, we recover the same regret of optimistic planning in standard MDPs. Also, as discussed in Line 278 - 279, the dependence on the observation rate is approximately $1 / \lambda_0^2$ (the square comes from consecutive observations for transition estimation).
>
> In Theorem 5.2, the dependence of the missing rate appears in the non dominating term. When the missing rate is 0, i.e., $v = \infty$, we can recover the regret bound in the standard MDPs with a slight modification on the proof similar response A3.
>
> >**Q6**: The executable policy class is defined based on the action sequence. However, based on the notation, the action is always known. Why is the action sequence $a_{t_h:h-1}$ important?
>
> **A6**: One has to keep track of past actions $a_{t_h:h-1}$ and choose actions based on them because they influence the current unknown state. It is necessary to augment state space with both past actions and the nearest observed state $s_{t_h}$, in order for the Markov property to hold and make the augmented MDP well defined. Missing any of $a_{t_h:h-1}$ would lead to incomplete information and ill-defined probability space.

---

> > ### Comment · Reviewer_yWj8 · 2023-08-16
> >
> > Thank you for the response and clarification.

---

> ### Comment · Area_Chair_kJEV · 2023-08-16
> **Are you satisfied by the answers?**
>
> Dear reviewer,
>
> Would you please indicate whether the authors' response is satisfactory for you? If not, please engage with the authors, so we can get a better assessment of this work.
>
> Thank you,
> Area Chair

---

### Author Rebuttal · Authors · 2023-08-10

We would like to thank all the reviewers for a thoughtful review and valuable comments, which help to improve the quality of the paper. We have provided our response to your individual questions and concerns.

In addition to clarify our contributions and technical novelties, we conduct numerical experiments as suggested by Reviewer kTAd. The results are summarized in the attached PDF file.

---

### Decision · Program_Chairs · 2023-09-21

**Decision:**

Accept (poster)

**Comment:**

The paper considers a modified MDP setting when either the observations are delayed or some of the observations are missing altogether. To solve this problem, it constructs an augmented MDP and proposes optimistic methods to solve them. The methods have regret guarantees, which in some cases is sharp. The paper also provides a result showing the gap between the optimal policy with no-delay MDP and with-delay MDP.

The reviewers are positive about this paper. We have two Borderline Accept, one Weak Accept, and One Accept. They believe that the problem is critical for applying RL to real-world scenarios, the theoretical analysis is well-executed, and the guarantees are sharp.

The reviewers mentioned a few shortcomings of the paper. Some of them are:

- The lack of experimental results (Reviewer kTAd).
The authors provided some preliminary empirical results during the rebuttal, so this is partially addressed.

- Difficulty of understanding how the augmented MDP is constructed (Reviewer 61zK).
I agree with this, and I hope the authors can provide some intuition about the construct. If the detail is not important for the main message of the paper, perhaps it can be postponed to an appendix.

- There has been concerns about the novelty of this work, especially interestingness of the algorithms, and the regret analysis following standard techniques or not (Reviewers yWj8 and 61zK).
The problem setup appears to be novel and is not reduced to any known setup. Naive approach does not get to the same regret. As these criticisms are more on the subjective side, I am not giving a significant weight to them.

- Missing steps in the the proof of Proposition 5.1 (Reviewer Mu3C).
I read this proof. It follows a pretty standard line of argument, though I suggest expanding it a bit to make it more accessible.

Overall, I believe **this is a technically good paper that should be accepted** at NeurIPS.